# Reward Learning as Doubly Nonparametric Bandits: Optimal Design and Scaling Laws

## Abstract

Specifying reward functions for complex tasks like object manipulation or driving is challenging to do by hand. Reward learning seeks to address this by learning a reward model using human feedback on selected query policies. This shifts the burden of reward specification to the optimal design of the queries. We propose a theoretical framework for studying reward learning and the associated optimal experiment design problem. Our framework models rewards and policies as nonparametric functions belonging to subsets of Reproducing Kernel Hilbert Spaces (RKHSs). The learner receives (noisy) oracle access to a true reward and must output a policy that performs well under the true reward. For this setting, we first derive non-asymptotic excess risk bounds for a simple plug-in estimator based on ridge regression. We then solve the query design problem by optimizing these risk bounds with respect to the choice of query set and obtain a finite sample statistical rate, which depends primarily on the eigenvalue spectrum of a certain linear operator on the RKHSs. Despite the generality of these results, our bounds are stronger than previous bounds developed for more specialized problems. We specifically show that the well-studied problem of Gaussian process (GP) bandit optimization is a special case of our framework, and that our bounds either improve or are competitive with known regret guarantees for the Matérn kernel.

## 1 Introduction

Specifying the reward function accurately for a desired objective, or *reward engineering*, is challenging to perform by hand, as the consequences of even small errors can be drastic (Hadfield-Menell et al., 2017). To address this, reward learning seeks to learn a predictive model of the reward function from data, which is obtained from carefully selected queries to human annotators. The learned reward model is then used as the optimization objective for policy learning. Reward learning has achieved significant empirical success in domains such as text summarization (Stiennon et al., 2020; Böhm et al., 2019), robot locomotion (Daniel et al., 2014), predicting driving styles (Kuderer et al., 2015), and Atari game playing (Christiano et al., 2017).

Despite their success, reward learning methods still lack theoretical grounding. Moreover, their behavior can be brittle even on simple tasks, due to the difficulty of choosing appropriate queries and due to feedback loops from adaptive querying (Freire et al., 2020). Indeed, an ablation study in Christiano et al. (2017) suggests that random queries can outperform or be competitive with adaptive query procedures. To address these issues, we provide a theoretical framework for analyzing reward learning, framing it as a *doubly nonparametric experimental design* problem. This framework helps elucidate the role of query selection (Chaloner & Verdinelli, 1995) and also enables us to derive scaling laws—how the sizes of the policy and reward models affect the query complexity—for reward learning (Kaplan et al., 2020).

**Proposed framework.** In our framework, we suppose we are given a reward class $C_r$ and policy class $C_\pi$. Our goal is to find a policy $\hat{\pi} \in C_\pi$ that performs well according to an unknown true reward $r^* \in C_r$. To do this, we query policies $\pi \in C_\pi$, observing noisy estimates of their true reward, and use this information to choose the eventual policy $\hat{\pi}$.

To be compatible with modern nonparametric learning methods (i.e. neural nets), we view $C_r$ and $C_\pi$ as subsets of Reproducing Kernel Hilbert Spaces (RKHS). The learner therefore optimizes a nonparametric reward function over a nonparametric space of policies, making the task "doubly" non-

parametric. In contrast, previous work considers a nonparametric function class or reward class, but typically not both. For instance, nonparametric zeroth order or bandit optimization (Srinivas et al., 2010; Mockus, 2012; Wang et al., 2018) considers a nonparametric function on a *finite-dimensional* input space. Conversely, nonparametric supervised learning (Wahba, 1990; Hofmann et al., 2008) minimizes a *known* loss function over a nonparametric input space.

The doubly nonparametric nature of our task poses new challenges. The (possibly) infinite-dimensional RKHS requires the learner to select which subspace to explore given a finite number of queries. Furthermore, the unknown reward function makes it challenging for the learner to reason about the information gained from the selected query policies. We address these challenges by deriving a risk upper bound for a family of plug-in estimators based on ridge regression, and then optimizing this bound to solve the optimal design task.

In addition to the optimal design problem, our framework allows us to study scaling laws with respect to the reward (or policy) class by varying the rate of decay of their corresponding eigen-spectrum. This decay rate determines the effective dimensionality of a RKHS (Zhang, 2002), and provides a natural proxy for varying the the size of the reward or policy class. Qualitatively, our main results show that the excess risk asymptotically vanishes as long as the policy class grows at a slower rate relative to the reward class.

**Sharpness of analysis.** Our risk bounds apply to reward and policy classes of arbitrary or even infinite dimensionality. Despite this generality, we show they provide stronger guarantees than previous known bounds for the specialized settings of compact policy sets and kernel multi-armed bandits.

In Section 4.3, we look at a special case of our problem when the policy set $C_\pi$ is a compact subspace and thus has finite rank. For these instances, we show that our learning algorithm obtains a better excess risk $O(n^{-\frac{\beta}{\beta+2}})$ versus a rate of $O(n^{-\frac{\beta-1}{2(\beta+1)}})$ obtained by the adaptive GP-UCB algorithm (Srinivas et al., 2010), where $\beta > 0$ is a power law decay rate.

In Section 5, we specialize our general results to the well-studied problem of Gaussian process bandit optimization (Williams & Rasmussen, 2006), also known as kernel multi-armed bandit (MAB). Specifically, for the class of Matérn kernels with parameter $\nu$ in $d$ dimensions, we show that our algorithm achieves a regret bound of $\tilde{O}(T^{\frac{4\nu+d(4d+6)}{6\nu+d(4d+7)}})$ which is strictly better than those achieved by the GP-UCB and GP-Thompson Sampling (GP-TS) (Chowdhury & Gopalan, 2017) algorithms and comparable with $\pi$-GP UCB (Janz et al., 2020) and supKernelUCB (Valko et al., 2013; Vakili et al., 2021); see Table 1 for details. GP-UCB and GP-TS are only yield sub-linear regret bounds when the smoothness of the kernel $\nu > d^2$—thus in high dimensions, these bounds essentially become vacuous. The $\pi$-GP UCB algorithm was designed specifically to overcome this issue. Our proposed algorithm achieves sublinear regret for all $\nu > 3/2$.

**Our Contributions.** We propose doubly-nonparametric bandits as a framework for theoretically studying the reward learning problem. Within this framework, we obtain finite sample risk bounds for a ridge regression based plug-in estimator and derive scaling laws for reward learning. From a technical standpoint, we study the optimal design problem for our estimator to select informative query points by showing that the excess risk depends only on the spectral properties of a certain operator of the two RKHSs and the empirical covariance matrix. As a corollary of our risk bounds, we provide sharper regret bounds for a class of kernel MAB problems compared to several existing algorithms, showing that the doubly-nonparametric lens of reward learning is fruitful even for "singly-nonparametric" tasks. To obtain these bounds, our reduction carefully constructs two different RKHSs to embed the input space and reward function into a policy and reward class.

## 2 FRAMEWORK: DOUBLY NONPARAMETRIC BANDITS

Our framework considers non-parametric policy learning with non-parametric reward models. We let $\pi \in \mathbb{H}_\pi$ denote an arbitrary policy and $r \in \mathbb{H}_r$ denote an arbitrary reward function, where $\mathbb{H}_\pi$ and $\mathbb{H}_r$ are Reproducing Kernel Hilbert Spaces. For technical reasons, we assume the corresponding kernel functions $\mathcal{K}_\pi$ and $\mathcal{K}_r$ both satisfy the Hilbert-Schmidt condition (see Appendix A for details).

We let $F(\pi, r) \in \mathbb{R}$ denote the reward obtained by selecting policy $\pi$ under reward function $r$ and consider the case where the evaluation functional $F$ is linear in both $\pi$ and $r$. In other words, $F(\pi, r) = \langle r, M\pi \rangle_{\mathbb{H}_r}$ where $M : \mathbb{H}_\pi \mapsto \mathbb{H}_r$ is a known linear mapping from the policy space to the

| Algorithm | Regret $\mathfrak{R}_T$ | Non-vacuous regime |
|---|---|---|
| GP-UCB (Srinivas et al., 2010) | $\tilde{O}(T^{\frac{2\nu+d(3d+3)}{4\nu+d(2d+2)}})$ | $\nu > \frac{d^2+d}{2}$ |
| GP-TS (Chowdhury & Gopalan, 2017) | $\tilde{O}(T^{\frac{2\nu+d(3d+3)}{4\nu+d(2d+2)}})$ | $\nu > \frac{d^2+d}{2}$ |
| Our work | $\tilde{O}(T^{\frac{4\nu+d(4d+6)}{6\nu+d(4d+7)}})$ | $\nu > \frac{3}{2}$ |
| $\pi$-GP-UCB (Janz et al., 2020) | $\tilde{O}(T^{\frac{2\nu+d(2d+3)}{4\nu+d(2d+4)}})$ | $\nu > 1$ |
| SupKernelUCB (Vakili et al., 2021) | $\tilde{O}(T^{\frac{\nu+d}{2\nu+d}})$ | $\nu > 1$ |

**Table 1.** Our algorithm specializes to the case of kernel multi-armed bandits and yields strong bounds (see eq. (9) for precise definition of regret). For a $d$-dimensional Matérn kernel with smoothness $\nu$, we outperform both GP-UCB and GP-TS unless $\nu \gtrsim d^2$. The only works to achieve better bounds for small $\nu$ are $\pi$-GP UCB, which was designed specifically for the Matérn kernel and a very recent analysis of the SupKernelUCB which achieves near minimax rates.

reward space. Since $\mathbb{H}_\pi$ and $\mathbb{H}_r$ may be infinite-dimensional, linearity is only a weak restriction–e.g. the map $f \mapsto f(x)$ is linear in $f$ for any RKHS.

To incorporate problem structure, we let $r^*$ denote the true reward function and assume that $r^* \in C_r$ for some known set $C_r \subseteq \mathbb{H}_r$ such that $\|r^*\|_{\mathbb{H}_r} = 1$. We further assume that policies $\pi$ are restricted to lie in some $C_\pi$ which is a subset of the unit ball in $\mathbb{H}_\pi$ (for instance, $C_\pi$ might incorporate physical constraints on implementable policies). Thus, given the true reward $r^*$, the optimal policy (for a compact $C_\pi$) is $\pi^* \in \mathrm{argmax}_{\pi \in C_\pi} F(\pi, r^*)$. This proposed framework, which allows for infinite-dimensional policy as well as reward classes, allows us to study how both the policy and reward space affect the difficulty of learning.

**Query access to reward $r^*$.** The true reward function $r^*$ is unknown to the learner but is accessible via queries to an oracle (e.g. a human expert), which provide noisy zeroth-order (or bandit) evaluations of the reward $r^*$. When queried with a policy $\pi \in C_\pi$, the oracle provides a response

$$\text{Oracle } \mathcal{O}_{r^*} : \pi \mapsto F(\pi, r^*) + \epsilon \quad \text{where} \quad \epsilon \sim \mathcal{N}(0, \tau^2) , \tag{1}$$

with $\tau^2$ denoting the variance of the response. There are two possible query models: passive queries (Atkinson, 1996; Sebastiani & Wynn, 2000), where the learner selects all queries at the same time, and active queries (Bubeck et al., 2011; Lattimore & Szepesvári, 2020), where the learner is allowed to select queries sequentially. Our focus in this work will be on the passive query model, but in many cases we will outperform existing active query algorithms.

**Problem statement.** Given passive access to the oracle $\mathcal{O}_{r^*}$, the objective of the learner is to output a policy $\hat{\pi} \in C_\pi$ that has small excess risk $\Delta$, defined as

$$\Delta(\hat{\pi}; r^*) := F(\pi^*, r^*) - F(\hat{\pi}, r^*) . \tag{2}$$

We think of queries to the oracle as expensive, and are interested in achieving low excess risk with as few queries as possible. This notion of excess risk is also studied by the term *simple regret* in pure exploration bandit problems (Lattimore & Szepesvári, 2020).

**Representations in $\ell_2(\mathbb{N})$.** By Mercer's theorem, we can represent any RKHS as a subset of $\ell_2(\mathbb{N})$. Formally, the policy and the reward spaces are isomorphic to the ellipsoids

$$\mathbb{H}_\pi := \left\{ \sum_{j=1}^\infty \kappa_{\pi,j} \phi_{\pi,j} \,\Big|\, (\kappa_{\pi,j})_{j=1}^\infty \in \ell^2(\mathbb{N}) \text{ with } \sum_{j=1}^\infty \frac{\kappa_{\pi,j}^2}{\mu_{\pi,j}^2} < \infty \right\} \text{ and}$$

$$\mathbb{H}_r := \left\{ \sum_{j=1}^\infty \kappa_{r,j} \phi_{r,j} \,\Big|\, (\kappa_{r,j})_{j=1}^\infty \in \ell^2(\mathbb{N}) \text{ with } \sum_{j=1}^\infty \frac{\kappa_{r,j}^2}{\mu_{r,j}^2} < \infty \right\} ,$$

for appropriately chosen eigenfunctions $\phi_{\pi,j}$ and $\phi_{r,j}$, and corresponding eigenvalues $\mu_{\pi,j}$ and $\mu_{r,j}$ (Wainwright, 2019). These are defined with respect to a base measure $\mathbb{P}$ over the input domain; see Appendix A for details. With a slight abuse of notation, going forward, we will use $\pi$ and $r$ to denote the corresponding coefficients $(\kappa_{\pi,j})$ and $(\kappa_{r,j})$ in the expansion above. [1] With this, the

---

[1]While the eigenfunctions $\phi_\pi$ and $\phi_r$ can be different, this representation can still be used by modifying the map $M$ appropriately. This is detailed in Appendix A.

---

**Algorithm 1:** Policy Learning via Reward Learning

---

**Input:** Number of queries $n$, policy set $C_\pi$, oracle $\mathcal{O}_{r^*}$

Select $n$ policies $\mathcal{Q} = \{\pi_1, \ldots, \pi_n\}$ and receive noisy reward evaluations $y_i = \mathcal{O}_{r^*}(\pi_i)$.

Estimate $\hat{r}$ using observed responses $\{(\pi_1, y_1), \ldots, (\pi_n, y_n)\}$ using ridge regression (4).

Obtain plug-in policy $\hat{\pi}_{\mathsf{plug}} \in \mathrm{argmax}_{\pi \in C_\pi} F(\pi, \hat{r})$.

**Output:** Policy $\hat{\pi}_{\mathsf{plug}}$

---

inner products associated with $\mathbb{H}_\pi$ and $\mathbb{H}_r$ simplify

$$\langle \pi_1, \pi_2 \rangle_{\mathbb{H}_\pi} := \sum_{j=1}^{\infty} \frac{\pi_{1,j} \pi_{2,j}}{\mu_{\pi,j}} \quad \text{and} \quad \langle r_1, r_2 \rangle_{\mathbb{H}_r} := \sum_{j=1}^{\infty} \frac{r_{1,j} r_{2,j}}{\mu_{r,j}} . \tag{3}$$

Also let $S_r := \mathrm{diag}(\mu_{r,j}^{-1})$ and $S_\pi := \mathrm{diag}(\mu_{\pi,j}^{-1})$ be diagonal matrices comprising the inverse of the eigenvalues of $\mathbb{H}_r$ and $\mathbb{H}_\pi$. With this notation, if we view the map $M$ as a (infinite-dimensional) matrix, its Hermitian adjoint[2] is equal to $M^* = S_\pi^{-1} M^\top S_r$.

In order for the evaluation functional $F(\pi, r^*)$ to be finite for all $\pi \in \mathbb{H}_\pi$, the operator norm $\|S_r^{\frac{1}{2}} M S_\pi^{-\frac{1}{2}}\|_{\mathsf{op}}$ must be bounded (see Appendix A). We will see later that the decay of this operator's singular values is closely related to the difficulty of learning in our setting.

# 3 ALGORITHM: POLICY LEARNING VIA REWARD LEARNING

Given the setup above, we now describe a meta-algorithm, policy learning via reward learning (Algorithm 1), for the non-parametric policy learning problem. The algorithm is a three-stage procedure: it (i) selects a subset of policies $\mathcal{Q}$ to query for reward feedback, (ii) uses the responses to learn a reward estimate $\hat{r}$, and (iii) optimizes this learnt estimate to output the policy $\hat{\pi}_{\mathsf{plug}}$, that is, $\hat{\pi}_{\mathsf{plug}} \in \mathrm{argmin}_{\pi \in C_\pi} \langle \hat{r}, M\pi \rangle_{\mathbb{H}_r}$. Such general plug-in procedure have been studied in the statistics (Van der Vaart, 2000) and the machine learning (Devroye et al., 2013) literature. We analyze the excess risk of this estimator for our doubly-nonparametric setup and use this risk bound to select our query set $\mathcal{Q}$. We now discuss the two key design choices in our algorithm: the choice of the reward estimation procedure as well as the choice of query set $\mathcal{Q}$.

**Reward learning via ridge regression.** We estimate the reward $\hat{r}$ via ridge regression in the RKHS $\mathbb{H}_r$ (Friedman et al., 2001; Shawe-Taylor et al., 2004). Suppose that in the first step of the algorithm, we have already queried the oracle on $n$ policies and let $\{(\pi_i, y_i)\}_{i=1}^n$ represent the query-response pairs. For a regularization parameter $\lambda_{\mathsf{reg}} > 0$, the ridge regression estimate of the reward function is

$$\hat{r} \in \mathrm{argmin}_{r \in \mathbb{H}_r} \frac{1}{n} \sum_{i=1}^{n} (y_i - \langle r, M\pi_i \rangle_{\mathbb{H}_r})^2 + \lambda_{\mathsf{reg}} \|r\|_{\mathbb{H}_r}^2 . \tag{4}$$

The parameter $\lambda_{\mathsf{reg}}$, which is usually set as a function of $n$, controls the bias-variance trade-off in estimating $r^*$—smaller values of $\lambda_{\mathsf{reg}}$ reduce bias while larger values help reduce variance.

**Excess risk bound for fixed query set.** Observe that the plug-in estimator $\hat{\pi}_{\mathsf{plug}}(\mathcal{Q})$ is implicitly a function of the query set $\mathcal{Q}$. Ideally, we want to choose the set $\mathcal{Q}$ which minimizes the expected risk of the plugin estimator. This requires us to solve the optimization problem

$$\mathcal{Q} = \mathrm{argmin}_{S:|S| \leq n} \mathbb{E}[\Delta(\hat{\pi}_{\mathsf{plug}}(S); r^*)] . \tag{5}$$

However, solving the above precisely requires knowledge about the underlying reward function $r^*$, and the combinatorial nature of the optimization problem makes it hard to find an exact solution. To address this, we first upper bound the excess risk of the plug-in policy $\hat{\pi}_{\mathsf{plug}}$ in terms of the query set $\mathcal{Q} = \{\pi_1, \ldots, \pi_n\}$. The following theorem[3] bounds the excess risk in terms of the spectrum of the spaces $\mathbb{H}_r$ and $\mathbb{H}_\pi$, as well as the covariance matrix of the queried policies $\Sigma_{\mathcal{Q}} := \frac{1}{n} \sum_{\pi \in \mathcal{Q}} \pi \pi^\top$.

---

[2]Recall the Hermitian adjoint of $M$ satisfies $\langle r, M\pi \rangle_{\mathbb{H}_r} = \langle M^* r, \pi \rangle_{\mathbb{H}_\pi}$

[3]Throughout the paper, for clarity purposes, we denote by $c$ a universal constant whose value changes across lines. All our proofs in the appendices explicitly track this constant.

**Theorem 1** (Excess risk of plug-in). *For any query set $\mathcal{Q}$ consisting of $n$ policies and regularization parameter $\lambda_{\text{reg}} > 0$, the excess risk of the plug-in estimator $\hat{\pi}_{\text{plug}}$ is upper bounded as*

$$\mathbb{E}[\Delta(\hat{\pi}_{\text{plug}}; r^*)] \leq 2\mathbb{E}[\|M^*(r^* - \hat{r})\|_{\mathbb{H}_\pi}] . \tag{6}$$

*In addition, letting $A = M\Sigma_{\mathcal{Q}}M^\top S_r + \lambda_{\text{reg}}I$, the expected squared distance is equal to*

$$\mathbb{E}[\|M^*(r^* - \hat{r})\|_{\mathbb{H}_\pi}^2] = \lambda_{\text{reg}}^2 \cdot \|M^* A^{-1} r^*\|_{\mathbb{H}_\pi}^2 + \frac{\tau^2}{n} \cdot \text{tr}\left[S_\pi(M^* A^{-1} M)\Sigma_{\mathcal{Q}}(M^* A^{-1} M)^\top\right]. \tag{7}$$

The proof follows a standard analysis of ridge regression and is deferred to Appendix B. Observe that in the above theorem, the query set $\pi \in \mathcal{Q}$ participates in the excess risk only via the covariance $\Sigma_{\mathcal{Q}}$. The risk bound is the sum of two term: the first corresponding to the bias and the second corresponding to the variance. In both these terms, $\Sigma_{\mathcal{Q}}$ appears as part of $A^{-1}$—thus query sets $\mathcal{Q}$ which induce a larger correlation with the map $M$ will generally have lower excess risk. Choices of queries which are orthogonal to the right singular vectors of $M$ will have a constant excess risk, since for those directions the matrix $A \approx \lambda_{\text{reg}}I$.

As shown later in the appendix, in the special case when the policy set consists of the entire unit ball $C_\pi = \{\pi \in \mathbb{H}_\pi \mid \|\pi\|_{\mathbb{H}_\pi} \leq 1\}$, the excess risk bound can be improved by a quadratic factor: $\mathbb{E}[\Delta(\hat{\pi}_{\text{plug}}; r^*)] \leq O\left(\|M^*(r^* - \hat{r})\|_{\mathbb{H}_\pi}^2\right)$. Such a phenomenon was first observed in the finite-dimensional setup by Rusmevichientong & Tsitsiklis (2010).

## 4 QUERY SELECTION AND STATISTICAL GUARANTEES

We now show how to select the query set $\mathcal{Q}$ effectively and study the excess risk of the corresponding plug-in estimator $\hat{\pi}_{\text{plug}}$. We will start with the special case where the policy set $C_\pi$ is the unit ball in $\mathbb{H}_\pi$ and the map $M$ is diagonal, and then generalize to arbitrary policy sets. In both cases, low excess risk can be achieved by repeatedly querying (approximations of) the projections of top eigenvectors of $M^*M$ onto the $\mathbb{H}_\pi$ space. For the special case when the map $M$ is diagonal, this is reduces to querying the top eigenvectors of $\mathbb{H}_\pi$.

The excess risk will ultimately depend on the the eigenspectrum of the operator $S_\pi^{-\frac{1}{2}} M^\top S_r M S_\pi^{-\frac{1}{2}}$, which is similar to the operator $M^*M$. Additionally, to interpret our results, we instantiate them for a power law spectrum with exponent $\beta > 0$, that is, $\sigma_j(S_\pi^{-\frac{1}{2}} M^\top S_r M S_\pi^{-\frac{1}{2}}) \asymp j^{-\beta}$. where $\sigma_j$ corresponds to the $j^{th}$ singular value of the corresponding operator. Such power law spectra have been observed in a variety of practical settings, for instance, in the Hessian of trained deep neural networks (Ghorbani et al., 2019).

### 4.1 WARM-UP: $C_\pi$ = UNIT BALL, $M$ = DIAGONAL

In order to get some intuition, we study the special case where the policy set $C_\pi$ consists of the entire unit ball in the space $\mathbb{H}_\pi$ and the map $M$ is diagonal with $M = \text{diag}(\nu_j)$. Further, let us denote the operator $\tilde{M} = S_r^{1/2} M S_\pi^{-1/2}$.

For this special case, our sampling algorithm (Algorithm 2) simply selects the top $J$ eigenvectors of the space $\mathbb{H}_\pi$ to query, for some value $J$ which depends on the decay exponent $\beta$. To see why, observe that for a diagonal map $M$, the right singular vectors of the operator $\tilde{M}$ are the same as the eigenvectors of the policy space $\mathbb{H}_\pi$. Therefore, the choice of policy $\pi_j$ in our algorithm is simply the scaled eigenfunction $\sqrt{\mu_{\pi,j}} \cdot \phi_{\pi,j}$. Having selected these $J$ queries, the algorithm queries each one of the $\frac{n}{J}$ times and uses this as query set $\mathcal{Q}$.

The intuition for this choice of query set $\mathcal{Q}$ is that since we are in the passive setup with no knowledge of $r^*$, any policy $\pi \in C_\pi$ can be an optimal policy. By querying the top $J$ ones out of these, we can obtain a good enough approximation to the performance of any policy in the unit ball. The particular choice of the parameter $J$ depends on the number of queries $n$ available. Since the oracle responses are noisy, to reduce variance in the responses along those directions, our algorithm performs multiple queries along the same direction.

If we further consider the special case when the policies and rewards correspond to the unit balls in the finite dimensional spaces $\mathbb{R}^{d_\pi}$ and $\mathbb{R}^{d_r}$ respectively, our choice of query set queries the directions

$\{e_i\}_{i=1}^{d_\pi}$, each for $J = \frac{n}{d_\pi}$ number of times. Intuitively, this strategy works well because without any prior over the unknown reward function, the optimal strategy in the passive setup is to explore all directions equally and this is precisely our set of chosen queries. This simple query strategy enjoys the following excess risk bound.

**Proposition 1** (Risk bound for $C_\pi$ = unit ball.). *For any $J \leq n$ and regularization parameter $\lambda_{\mathsf{reg}} > 0$, consider the plug-in estimator obtained via the passive sampling algorithm which explores the first $J$ eigenfunctions of $\mathbb{H}_\pi$. The excess risk satisfies*

$$\mathbb{E}[\Delta(\hat{\pi}_{\mathsf{plug}}; r^*)] \leq c \cdot \left(1 + \frac{\tau^2}{n\lambda_{\mathsf{reg}}^2}\right) \cdot \max\left\{\sup_{j \leq J} \frac{\lambda_{\mathsf{reg}}^2 J^2 \zeta_j}{\zeta_j^2 + \lambda_{\mathsf{reg}}^2 J^2}, \sup_{j > J} \zeta_j\right\} , \qquad (8)$$

*where the quantity $\zeta_j = \frac{\nu_j^2 \mu_{\pi,j}}{\mu_{r,j}}$ and $c > 0$ is some universal constant.*

We defer the proof of the above proposition to Appendix B. The choice of the exploration parameter $J$ allows us to trade-off between the two terms inside the maximum. Typically, the second term will be maximized at $j = J + 1$. For the first term, the supremum depends on the choice of $\lambda_{\mathsf{reg}}$ — for small values of $\lambda_{\mathsf{reg}}$, the sup is achieved at $j = 1$ while for larger values, it is achieved at $j = J$. In order to gain more intuition about this bound, we instantiate this for the power law decay.

**Corollary 1** (Risk bound for power-law decay). *Suppose that eigenvalues of the police space $\mathbb{H}_\pi$ decay as $j^{-\beta_\pi}$, reward space $\mathbb{H}_r$ as $j^{-\beta_r}$ and the singular values of map $M$ as $j^{-\beta_M}$. This satisfies the power law assumption with exponent $\beta = \beta_\pi + \beta_M - \beta_r$. The plug-in estimator with exploration parameter $J = n^{\frac{1}{\beta+2}}$ and regularization $\lambda_{\mathsf{reg}} = n^{-\frac{\beta+1}{\beta+2}}$ satisfies $\mathbb{E}[\Delta(\hat{\pi}_{\mathsf{plug}}; r^*)] \leq cn^{-\frac{\beta}{\beta+2}}$.*

The above bound shows that our algorithm can learn in the framework as long as $\beta > 0$ or equivalently $\beta_\pi + \beta_M > \beta_r$, with better rates for larger values of $\beta$. Thus, for a fixed size of reward class $\beta_r$, the learning rate improves as the policy class grows smaller ($\beta_\pi$ increases) – this is intuitive since we are required to search over a smaller policy space. On the other hand, for a fixed policy class $\beta_\pi$, our excess risk rate gets better as the reward class grows in size ($\beta_r$ increases) – this is because a larger set of reward functions have similar optimal policies and hence learning gets easier.

## 4.2 GENERAL POLICY SETS

We now describe our choice of query sets $\mathcal{Q}$ for general policy sets $C_\pi$. Our strategy, described in Algorithm 2, differs from the above special case in that we need to take into account the interaction of the policy space $\mathbb{H}_\pi$ with the map $M$. Specifically, we show in Appendix B that the upper bound in Theorem 1 can be diagonalized for this general case via a transformation.

Let us denote the operator $\tilde{M} = S_r^{1/2} M S_\pi^{-1/2}$. Our transformation reveals that the relevant directions to query for this general case corresponds to the columns of $\Phi_\pi S_\pi^{-1/2} \Phi_\pi^\top V_M$ where , then $V_M$ are the eigenvectors of the self-adjoint operator $\tilde{M}^\top \tilde{M}$ – and it is precisely a subset of these directions that our algorithm queries.

In order to be able to query these policies, we require the set $C_\pi$ to contain some policies which align well with them. We formally state this regularity assumption below.

**Assumption 1** (Regularity assumption on $C_\pi$). *For any eigenfunction $\phi_{\tilde{M},j}$ of the operator $\tilde{M}^\top \tilde{M}$, consider the policy $\pi_j = \Phi_\pi S_\pi^{-1/2} \Phi_\pi^\top \phi_{\tilde{M},j}$. There exists a policy $\tilde{\pi}_j$ in policy set $C_\pi$ such that for some constant $c_\pi > 0$, we have $\tilde{\pi}_j \tilde{\pi}_j^\top \succeq c_\pi \pi_j \pi_j^\top$.*

The above assumption requires that for every choice of the policy $\pi_j$ in Algorithm 2, the set $C_\pi$ has the another policy $\tilde{\pi}_j$ which is collinear with it. This assumption can be relaxed in various ways (for instance via convexification) but we omit this as it is not needed for our results. Given this assumption, the following theorem, a generalization of Proposition 1, provides a bound on the excess risk for the plug-in estimate for general policy sets $C_\pi$.

**Theorem 2** (Risk bound for general policy sets $C_\pi$.). *For any $J \leq n$, regularization parameter $\lambda_{\mathsf{reg}} > 0$ and set $C_\pi$ satisfying Assumption 1, let $\hat{\pi}_{\mathsf{plug}}$ be the estimator output by Algorithm 1. The squared excess risk satisfies*

$$(\mathbb{E}[\Delta(\hat{\pi}_{\mathsf{plug}}; r^*)])^2 \leq c \cdot \left(1 + \frac{\tau^2}{n\lambda_{\mathsf{reg}}^2}\right) \cdot \max\left\{\sup_{j \leq J} \frac{\lambda_{\mathsf{reg}}^2 J^2 \zeta_j}{\zeta_j^2 + \lambda_{\mathsf{reg}}^2 J^2}, \sup_{j > J} \zeta_j\right\} ,$$

---

**Algorithm 2:** Passive querying strategy

---

**Input:** Number of queries $n$, map $M$, policy set $C_\pi$, exploration parameter $J$

Construct linear map $\tilde{M} = S_r^{\frac{1}{2}} M S_\pi^{-\frac{1}{2}}$ and compute eigenvectors $\{\phi_{\tilde{M},j}\}_j$ of operator $\tilde{M}^\top \tilde{M}$

Set policy $\pi_j = \Phi_\pi S_\pi^{-\frac{1}{2}} \Phi_\pi^\top \phi_{\tilde{M},j}$ for all $j \leq J$

Obtain policy $\tilde{\pi}_j \in C_\pi$ such that $\tilde{\pi}_j \tilde{\pi}_j^\top \succeq c_\pi \pi_j \pi_j^\top$

Form query set $\mathcal{Q} = \{\tilde{\pi}_1^{(n/J)}, \ldots, \tilde{\pi}_{n^\alpha}^{(n/J)}\}$ where $a^{(b)} = \{a, \ldots, a\}\}$ repeated $b$ times

**Output:** Query set $\mathcal{Q}$

---

*where the values $\zeta_j$ correspond to the $j^{th}$ eigen values of the operator $\tilde{M}^* \tilde{M}$ with $\tilde{M} = S_r^{\frac{1}{2}} M S_\pi^{\frac{1}{2}}$.*

We defer the proof of this theorem to Appendix B. The proof of this theorem goes via a transformation which diagonalizes the excess risk bound and reduces the problem to a similar setup as that of Proposition 1. Additionally, Assumption 1 allows us to generalize the results to arbitrary policy sets $C_\pi$. Note that the above upper bounds the square of the excess risk. As discussed in Section 3, one can obtain a quadratic improvement in this rate if the set $C_\pi$ is the entire unit ball in $\mathbb{H}_\pi$. We specialize the above bound for the power law decay assumption in the following corollary.

**Corollary 2** (Risk bound for power-law decay)**.** *Suppose that eigenspectrum of the operator $S_\pi^{-\frac{1}{2}} M^\top S_r M S_\pi^{-\frac{1}{2}}$ satisfy the power law assumption with exponent $\beta > 0$, that is, $\sigma_j(S_\pi^{-\frac{1}{2}} M^\top S_r M S_\pi^{-\frac{1}{2}}) \asymp j^{-\beta}$. The plug-in estimator $\hat{\pi}_{\mathsf{plug}}$ with parameter $J = n^{\frac{1}{\beta+2}}$ and regularization $\lambda_{\mathsf{reg}} = n^{-\frac{\beta+1}{\beta+2}}$ satisfies $\mathbb{E}[\Delta(\hat{\pi}_{\mathsf{plug}}; r^*)] \leq c n^{-\frac{\beta}{2(\beta+2)}}$.*

The above bound indicates that for the general case, learning is possible if the spectrum decay has parameter $\beta > 0$. To get such a spectrum decay with the operator defined in the above corollary, one sufficient condition is that the map $M$ does not flip the larger eigenvectors of $\mathbb{H}_\pi$ towards the smaller eigenvectors of $\mathbb{H}_r$, that is, the map $M$ preserves the ordering of the eigenvectors of $\mathbb{H}_\pi$ when transformed to the space $\mathbb{H}_r$. Such a misaligned scenario would require learning a very accurate representation of the reward to learn a good policy and will make learning harder.

### 4.3 COMPARISON WITH UCB-STYLE ADAPTIVE ALGORITHMS

We next turn to evaluating the sharpness of Theorem 2. Existing frameworks for studying "singly"-nonparametric setups require the input domain to be compact. In our doubly-nonparametric setup, the input space is the policy set $C_\pi$ which is often non-compact (i.e. the unit ball is not compact in infinite dimensions). We address this for singly-nonparametrics algorithm by taking a finite-dimensional approximation.

Even though our proposed method is passive, it achieves better rates than well-known *adaptive* sampling algorithms. Specifically, in the power law setting of Section 4.1, the analysis of GP-UCB algorithm (Srinivas et al., 2010) provides a rate of $O(n^{-\frac{\beta-1}{2(\beta+1)}})$, which is strictly worse than the $O(n^{-\frac{\beta}{\beta+1}})$ obtained by our analysis in Corollary 1. We refer the reader to Proposition 2 in Appendix D for an exact statement. The proof adapts the analysis from Srinivas et al. (2010), which hinges on a quantity called the information gain, which we bound for our setup. While we are comparing upper bounds for the two algorithms, we believe that our improved bound is due to a better algorithm and not an analysis gap. While we expect adaptive algorithms to perform better than passive ones in general (Lattimore & Hao, 2021), UCB style algorithms require the construction of confidence intervals around input points, which crucially dictate the regret bounds of such algorithms. In the frequentist setup, the best known such bounds (Vakili et al., 2021) are known to yield suboptimal regret rates and it is an open question as to whether these can be improved.

## 5 NEW BOUNDS FOR KERNEL MULTI-ARMED BANDITS

In the previous subsection, we saw that our passive sampling algorithm actually outperforms existing adaptive sampling algorithms for the reward learning task we care about. Here we take this a step

further—we specialize our algorithm to the case of kernel MABs, and show that it outperforms standard algorithms for that setting and is competitive with a specialized algorithm for Matérn kernels.

We consider the task of maximizing an unknown function $f^* : \mathcal{X} \mapsto \mathbb{R}$ over its domain $\mathcal{X} \subset \mathbb{R}^d$. In the kernel multi-armed bandit (MAB) setup, this unknown function $f$ belongs to an RKHS $\mathbb{H}$, equipped with a positive-definite kernel[4] $\mathcal{K}$, such that $\|f^*\|_{\mathbb{H}} = 1$. Let us further restrict our attention to the space of input points $\mathcal{X} = \{x \in \mathbb{R}^d \mid \|x\|_2 \leq 1\}$. The learner is allowed to access this function via a noisy zeroth-order oracle $\mathcal{O}_{f^*} : x \mapsto f^*(x) + \eta$ where $\eta \sim \mathcal{N}(0, \tau^2)$. Going forward we will assume that $\tau = 1$. The above oracle is similar to the reward oracle $\mathcal{O}_{r^*}$, except that the query points $x$ belong to a finite dimensional space and $f^*$ is a non-linear function of the query point $x$. The goal in MAB is to minimize the $T$-step regret

$$\mathfrak{R}_T := \max_{x \in \mathcal{X}} f^*(x) - \sum_{t=1}^{T} f^*(x_t) , \tag{9}$$

where $x_t$ is the datapoint queried in the $t^{th}$ round. There have been several algorithms proposed to solve this problem including general purpose UCB algorithms (Srinivas et al., 2010; Chowdhury & Gopalan, 2017), Thompson sampling approaches (Chowdhury & Gopalan, 2017), and special-purpose algorithms for specific kernels (Janz et al., 2020).

We next show that kernel MAB can be cast as a special case of our non-parametric policy learning framework. The resulting regret bounds, derived from an application of Theorem 3, are better than several general purpose algorithms (GP-UCB, IGP-UCB, GP-TS) and comparable to those specialized for the Matérn kernel ($\pi$-GP-UCB).

In order to reduce kernel MAB to our framework, we need to introduce three elements – the policy space $\mathbb{H}_\pi$, the reward space $\mathbb{H}_r$ and the map $M$. We would like spaces $\mathbb{H}_r$ and $\mathbb{H}_\pi$ such that (1) the resulting objective $F(r, \pi)$ is linear in this space, (2) the resulting rewards and policies have unit norm in their respective space, and (3) we have a good understanding of the eigenvalues of the resulting operator. This last point ensures that we can employ our upper bounds from Section 4.

Before we define these, we let $\mathcal{C}_\epsilon$ denote an $\epsilon$-net of the input space $\mathcal{X}$ under the $\ell_2$ norm and denote its size by $N_{\mathsf{cov}}(\epsilon)$. We define the kernel matrix $K \in \mathbb{R}^{N_{\mathsf{cov}} \times N_{\mathsf{cov}}}$ on points selected in the cover as $K(i, j) = \mathcal{K}(x_i, x_j)$ for all $(x_i, x_j) \in \mathcal{C}_\epsilon \times \mathcal{C}_\epsilon$.

**Reward space $\mathbb{H}_r$.** Given the RKHS $\mathbb{H}$ as well as the elements of the cover $\mathcal{C}_\epsilon$, we view the reward function as a map from $\mathcal{C}_\epsilon$ to $\mathbb{R}$, or equivalently as a vector in $\mathbb{R}^{N_{\mathsf{cov}}(\epsilon)}$. More precisely, letting $\tilde{f} = [f(x_1), \ldots, f(x_{N_{\mathsf{cov}}})]$ denote the vector of evaluations of a function $f$, we define

$$\mathbb{H}_r := \mathrm{span}\{\tilde{f} \mid f \in \mathbb{H}\} \quad \text{with} \quad \langle \tilde{f}_1, \tilde{f}_2 \rangle_{\mathbb{H}_r} := \tilde{f}_1^\top K^{-1} \tilde{f}_2, . \tag{10}$$

With this notation, we define the true reward $r^* := \tilde{f}^* = [f^*(x_1), \ldots, f^*(x_{N_{\mathsf{cov}}})]$.

**Policy Space $\mathbb{H}_\pi$.** Similarly to rewards, we will embed policies in $\mathbb{R}^{N_{\mathsf{cov}}}$. For any point $x \in \mathcal{C}_\epsilon$, let $k_x = [\mathcal{K}(x, x_1), \ldots, \mathcal{K}(x, x_{N_{\mathsf{cov}}})]$ denote the corresponding vector in $\mathbb{R}^{N_{\mathsf{cov}}}$ obtained by evaluating the kernel $\mathcal{K}$ over the cover. Then, the space

$$\mathbb{H}_\pi := \mathrm{span}\{k_x \mid x \in \mathcal{C}_\epsilon\} \quad \text{with} \quad \langle k_1, k_2 \rangle_{\mathbb{H}_\pi} := \langle k_1, K^{-2} k_2 \rangle . \tag{11}$$

The choice of the above norm ensures that $\langle k_i, k_j \rangle_{\mathbb{H}_\pi} = \langle K^{-1} k_i, K^{-1} k_j \rangle = \delta_{i,j}$ for all $(x_i, x_j) \in \mathcal{C}_\epsilon \times \mathcal{C}_\epsilon$ . Thus in particular, $\mathbb{H}_\pi$ contains an orthonormal embedding of the set of vectors $\{k_x\}_{x \in \mathcal{C}_\epsilon}$.

**Map $M$.** Both the reward space $\mathbb{H}_r$ and policy space $\mathbb{H}_\pi$ can be associated with $\mathbb{R}^{N_{\mathsf{cov}}}$. Under this transformation, the evaluation $f^*(x)$ for any $x \in \mathcal{C}_\epsilon$ corresponds to the standard inner product with $F(r^*, \pi_x) = f^*(x) = (\tilde{f}^*)^\top K^{-1} k_x = \langle r^*, k_x \rangle_{\mathbb{H}_r}$. This indicates that we should take the map $M$ to be the identity. Furthermore, as a simple application of Mercer's theorem it follows that this map $M$ is a bounded linear operator.

We make an additional assumption on the kernel function $\mathcal{K}$, requiring it to be Lipschitz in its input arguments. This assumption is often satisfied, in particular for the Matérn kernel when $\nu > 3/2$.

**Assumption 2** (Lipschitz Kernel $\mathcal{K}$)**.** *The Kernel $\mathcal{K}$ associated with the Hilbert space $\mathbb{H}$ is $L_\mathcal{K}$-Lipschitz with respect to the $\ell_2$-norm for some $L_\mathcal{K} > 0$: $|\mathcal{K}(x, y) - \mathcal{K}(x, x)| \leq L_\mathcal{K}\|x - y\|_2$ for all $x \in \mathcal{X}, y \in \mathcal{X}$. Furthermore, the kernel satisfies $\mathcal{K}(x, x) = 1$ for all points $x \in \mathcal{X}$.*

---

[4]We require that the kernel $\mathcal{K}$ be a Mercer's kernel satisfying $\mathcal{K}(x, x) = c$ for all $x \in \mathcal{X}$.

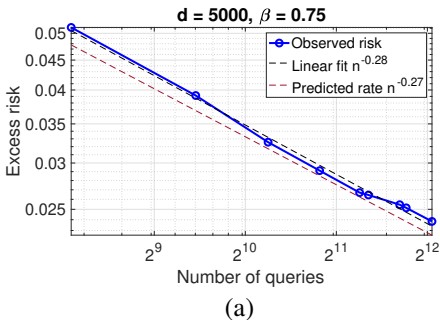 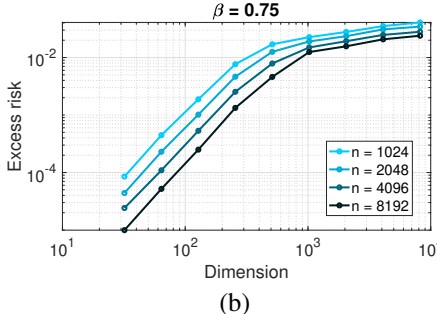

**Figure 1.** (a) Corroborating upper bound from Corollary 1. Our theoretical bounds predict a rate of $n^{-0.27}$ and the experiment shows an almost matching rate of $n^{-0.28}$. (b) As the dimension $d$ is increased, the excess risk curves asymptote at different levels for different $n$. This shows that our algorithm achieves non-vacuous error for the doubly-nonparametric set in the regime $d \to \infty$.

Applying Theorem 2 under the above assumption, we obtain the following excess risk bound for the plug-in estimator evaluated on the unknown function $f^*$.

**Theorem 3** (Excess risk for Kernel MAB). *Suppose that the eigenvalues of a $L_{\mathcal{K}}$-Lipschitz kernel $\mathcal{K}$ satisfy the power-law decay $\mu_j \asymp j^{-\beta}$. Let $\hat{x}_{\mathsf{plug}}$ be the output of Algorithm 1 using $n$ queries to the oracle $\mathcal{O}_{f^*}$. Then, for any value of $\beta > 1 + \frac{2}{d} + \log(\frac{1}{\delta})$ and $\epsilon \in (0,1)$, the excess risk satisfies*

$$\max_{x:\|x\|_2 \leq 1} f^*(x) - f^*(\hat{x}_{\mathsf{plug}}) \lesssim N_{\mathsf{cov}}^{\frac{1}{\beta+2}}(\epsilon) \cdot n^{\frac{-\beta}{2(\beta+2)}} + N_{\mathsf{cov}}^{\frac{1-\beta}{2}}(\epsilon) + \sqrt{L_{\mathcal{K}}\epsilon} \,,$$

*with probability at least $1 - \delta$.*

For Matérn kernels, it is known that the eigenvalues decay with parameter $\beta = 1 + \frac{2\nu}{d}$ (Janz et al., 2020). Using this, we can obtain the following corollary.

**Corollary 3** (Regret bound for Matérn Kernel). *Consider the family of Matérn kernels with parameter $\nu > \frac{3}{2}$ defined with the Euclidean norm over $\mathbb{R}^d$. The $T$-step regret of our algorithm is* $\mathfrak{R}_{\mathsf{mat},T} = \tilde{O}\left(T^{\frac{4\nu+d(6+4d)}{6\nu+d(7+4d)}}\right)$.

The above bound is for regret, which is an online notion, while our previous results are offline notions. We get from one to the other using a standard batch-to-online conversion bound based on an explore-then-commit strategy. Table 1 compares the above bound to the existing bounds in the literature. While the bounds for GP-UCB and GP-TS become vacuous for $\nu \lesssim d^2$, our bound from Corollary 3 is always sublinear in $T$.

## 6 EXPERIMENTAL EVALUATION

We experimentally evaluate our algorithm via a simulation study. We use these experiments to establish the dimension free nature of our results as well as to conjecture optimality of our bounds.

**Setup.** In the simulation study, we work with $d$ dimensional RKHSs $\mathbb{H}_r$ and $\mathbb{H}_\pi$. In order to simulate the nonparmeteric regime, we typically use value of $n$ which are less or at most a constant times the dimension $d$. We set the matrices $S_\pi = \mathrm{diag}(j^{-1.75})$, $S_r = \mathrm{diag}(j^{-1})$ and the map $M = I$. With this, the effective decay parameter $\beta = \beta_\pi - \beta_r = 0.75$. We further sampled the oracle noise $\epsilon \sim \mathcal{N}(0, 0.01)$. All plots were averaged over 10 runs.

**Observations.** Figure 1(a) shows the variation of excess risk as the number of queries $n$ are varied from 256 to 4096 on a log-log plot. Our bounds in Corollary 1 for this setup predict that the excess risk should decay at a rate $O(n^{-0.27})$. By fitting a linear line through the plot, we found that observed risk to vary as $O(n^{-0.28})$. This plot is suggestive of the fact that our theoretical upper bounds might be tight in a minimax way over choices of decay parameter $\beta$. In Figure 1(b), we plot the excess risk as we vary the dimension $d$ from 32 to 8192 for four different choices of sample size, again, on a log-log scale. Increasing the number of queries decreases the excess risk for all dimensions consistently. The risk curves tend to asymptote at different error levels for different values of $n$. This corroborates our theoretical findings that our proposed algorithm provides non-vacuous bounds for the doubly-nonparametric setup in the regime $d \to \infty$.

ETHICS STATEMENT

Our main contribution is a theoretical framework to study reward learning and the associated optimal design problem. Since our contributions are mostly theoretical in nature, we do not anticipate any ethical issues arising in the near future.

REPRODUCIBILITY STATEMENT

On the theoretical side, we provide detailed proofs for all our results in the appendix and appropriately reference the intermediate results we might have used in the proofs. For the experimental aspect, we have attached our matlab code as a supplementary file and have provided all necessary hyper parameters details required to reproduce the experiments.

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

## A  TECHNICAL DETAILS FOR PROPOSED FRAMEWORK

### A.1  RKHS ASSUMPTION

The Hilbert spaces $\mathbb{H}_\pi$ and $\mathbb{H}_r$ are Reproducing Kernel Hilbert Spaces defined by kernel functions $\mathcal{K}_\pi, \mathcal{K}_r : \mathcal{X} \times \mathcal{X} \mapsto [0, 1]$ respectively defined over a compact instance space $\mathcal{X}$. Further, the kernels $\mathcal{K}_\pi$ and $\mathcal{K}_r$ satisfy the Hilbert-Schmidt condition

$$\int_{\mathcal{X} \times \mathcal{X}} \mathcal{K}_i(x, z)^2 d\mathbb{P}(x) d\mathbb{P}(z) \leq \infty \quad \text{for } i = \{\pi, r\} \,, \tag{12}$$

for some distribution $\mathbb{P}$ over space $\mathcal{X}$. Mercer's theorem (Mercer, 1909) implies that such kernel functions have an associated set of eigenfunctions (with corresponding eigenvalues) that form an orthonormal basis for $L^2(\mathcal{X}, \mathbb{P})$. We restate a version of this theorem below (Wainwright, 2019).

**Theorem 4** (Mercer's theorem). *Suppose that the space $\mathcal{X}$ is compact and the positive semi-definite kernel $\mathcal{K}$ satisfies the Hilbert-Schmidt condition* (12)*. Then there exists a sequence of eigenfunctions $(\phi_j)_{j=1}^\infty$ that form an orthonormal basis of $L^2(\mathcal{X}, \mathbb{P})$ and non-negative eigenvalues $(\mu_j)_{j=1}^\infty$ such that*

$$\int_{\mathcal{X}} \mathcal{K}(x, z)\phi_j(z) d\mathbb{P}(z) = \mu_j \phi_j(x) \quad \text{for all } j = 1, 2, \ldots. \tag{13}$$

*Furthermore, the kernel function has the expansion*

$$\mathcal{K}(x, z) = \sum_{j=1}^\infty \mu_j \phi_j(x)\phi_j(z) \,, \tag{14}$$

*where the convergence of the sequence holds absolutely and uniformly.*

### A.2  CONDITIONS FOR REWARD BOUNDEDNESS

For learning to be feasible in the proposed framework, we would require that the evaluation functional $F(\pi, r^*)$ is bounded for any policy $\pi \in \mathbb{H}_\pi$. Using the fact that $\|r^*\|_{\mathbb{H}_r} \leq 1$ and $\|\pi\|_{\mathbb{H}_\pi} \leq 1$, we have

$$F(\pi, r^*) = \langle r^*, M\pi \rangle_{\mathbb{H}_r} = (r^*)^\top S_r M\pi \leq \|S_r^{\frac{1}{2}} M S_\pi^{-\frac{1}{2}}\|_{\text{op}} \,. \tag{15}$$

Thus one sufficient condition for the reward functional to be bounded is to ensure that the operator norm $\|S_r^{\frac{1}{2}} M S_\pi^{-\frac{1}{2}}\|_{\text{op}}$ is finite. In the special case when the map is diagonal with $M = \text{diag}(\nu_j)$, the above condition simplifies to

$$F(\pi, r^*) \leq \sup_{j \geq 1} \left[ \frac{\nu_j \mu_{\pi, j}^{\frac{1}{2}}}{\mu_{r, j}^{\frac{1}{2}}} \right] \,. \tag{16}$$

### A.3  REGULARITY ASSUMPTIONS ON MAP $M$

We assume that the map $M$ is a compact bounded operator from the policy space $\mathbb{H}_\pi$ to the reward space $\mathbb{H}_r$. By Schauder's theorem, the adjoint $M^*$ is also a compact operator. Thus, the map $M^*M : \mathbb{H}_\pi \to \mathbb{H}_\pi$ is a compact self-adjoint operator. This allows us to use the spectral theorem for compact self-adjoint operators which guarantees the existence of eignevalues and eignefunctions for the operator $M^*M$ and a corresponding singular value decomposition for the map $M$ (Kreyszig, 1978).

### A.4  NON-ALIGNED RKHSs

As mentioned in the Section 2, if the eigenvectors of the spaces $\mathbb{H}_r$ and $\mathbb{H}_\pi$ are not aligned, one can consider the following simple transformation which resolves this. Let $\Phi_\pi$ and $\Phi_r$ represent the eigenvectors.

$$\tilde{r} = \Phi_r r, \quad \tilde{\pi} = \Phi_\pi^\top \pi, \quad \text{and} \quad \tilde{M} = \Phi_r^\top M \Phi_\pi \,. \tag{17}$$

The above transformation implies that $\|\tilde{r}\|_{\mathbb{H}_r} \leq 1$ and $\|\tilde{\pi}\|_{\mathbb{H}_\pi} \leq 1$.

# B  PROOF OF MAIN RESULTS

In this section we provide the proofs for the main results of this work. Appendix D to follow contains the proofs for the other results.

## B.1  PROOF OF THEOREM 1

We begin by proving the result for the special case when the policy set $C_\pi$ consists of the entire unit ball and then generalize the analysis to arbitrary policy sets.

**Case 1: $C_\pi$ is unit ball in $\mathbb{H}_\pi$.**  For this special case, observe that the the optimal policy $\pi^*$ and the plug-in policy $\hat{\pi}_{\mathsf{plug}}$ for any reward estimate $\hat{r}$ can be written as

$$\pi^* = \frac{M^* r^*}{\|M^* r^*\|_{\mathbb{H}_\pi}} \quad \text{and} \quad \hat{\pi}_{\mathsf{plug}} = \frac{M^* \hat{r}}{\|M^* \hat{r}\|_{\mathbb{H}_\pi}} , \tag{18}$$

where the operator $M^*$ is the adjoint of of the map $M$. To prove a bound on the excess risk using the plug-in estimate, we use the following lemma which bounds this error in terms a deviation of the estimated and true rewards.

**Lemma 1.** *Consider any vectors $x$ and $y$ with finite non-zero norm under some inner product $\langle \cdot, \cdot \rangle$. Then, we have*

$$\langle x, \frac{x}{\|x\|} - \frac{y}{\|y\|} \rangle \leq \frac{\|x-y\|^2}{2\|y\|} . \tag{19}$$

The proof of the above lemma is presented in Section B.1.1. Taking the above as given, we can upper bound the excess risk

$$\begin{aligned}
\Delta(\hat{\pi}; r^*) &= \langle M^* r^*, \frac{M^* r^*}{\|M^* r^*\|_{\mathbb{H}_\pi}} - \frac{M^* \hat{r}}{\|M^* \hat{r}\|_{\mathbb{H}_\pi}} \rangle_{\mathbb{H}_\pi} \\
&\leq \frac{\|M^*(r^* - \hat{r})\|_{\mathbb{H}_\pi}^2}{2\|M^* \hat{r}\|_{\mathbb{H}_\pi}} .
\end{aligned} \tag{20}$$

**Case 2: Arbitrary set $C_\pi$.**  For this case, consider the excess risk of plug-in estimator $\hat{\pi}_{\mathsf{plug}}$ obtained by maximizing reward estimate $\hat{r}$

$$\begin{aligned}
\Delta(\hat{\pi}; r^*) &= \langle M^* r^*, \pi^* - \hat{\pi}_{\mathsf{plug}} \rangle_{\mathbb{H}_\pi} \\
&= \langle M^*(r^* - \hat{r}), \pi^* \rangle_{\mathbb{H}_\pi} + \langle M^* \hat{r}, \pi^* - \hat{\pi}_{\mathsf{plug}} \rangle_{\mathbb{H}_\pi} + \langle M^*(\hat{r} - r^*), \hat{\pi}_{\mathsf{plug}} \rangle_{\mathbb{H}_\pi} \\
&\overset{(i)}{\leq} 2\|M^*(r^* - \hat{r})\|_{\mathbb{H}_\pi} ,
\end{aligned} \tag{21}$$

where the final inequality follows from the fact that $\hat{\pi}_{\mathsf{plug}}$ maximizes $F(\pi; \hat{r})$ over the set $C_\pi$.

Thus, we see that for both the cases above, we can upper bound the excess risk of the plug-in estimator in terms of the norm $\|M^*(r^* - \hat{r})\|_{\mathbb{H}_\pi}$. Next, we evaluate this for the ridge regression based reward estimator for any set of $n$ queries $\mathcal{Q} = \{\pi_1, \ldots, \pi_n\}$ with covariance matrix $\Sigma = \frac{1}{n} \sum_i \pi_i \pi_i^\top$. For any regularization parameter $\lambda_{\mathsf{reg}} > 0$, we have,

$$\begin{aligned}
\hat{r} &= \arg \min_{r \in \mathbb{H}_r} \frac{1}{n} \sum_{i=1}^n (y_i - \langle r, M\pi_i \rangle_{\mathbb{H}_r})^2 + \lambda_{\mathsf{reg}} \|r\|_{\mathbb{H}_r}^2 \\
&\overset{(i)}{=} (M\Sigma M^\top S_r + \lambda_{\mathsf{reg}} I)^{-1} \cdot \frac{1}{n} \sum_{i=1}^n y_i M\pi_i \\
&= r^* - \lambda_{\mathsf{reg}} (M\Sigma M^\top S_r + \lambda_{\mathsf{reg}} I)^{-1} r^* + (M\Sigma M^\top S_r + \lambda_{\mathsf{reg}} I)^{-1} \left( \frac{M}{n} \sum_{i=1}^n \epsilon_i \pi_i \right),
\end{aligned} \tag{22}$$

where and equality (i) follows by substituting the value of $y_i = F(\pi_i, r^*) + \epsilon_i$. Let us denote by matrix $A = M\Sigma M^\top S_r + \lambda_{\text{reg}} I$. Therefore, the error in reward estimation

$$\hat{r} - r^* = \lambda_{\text{reg}} A^{-1} r^* + A^{-1} \left( \frac{M}{n} \sum_{i=1}^{n} \epsilon_i \pi_i \right)$$

$$\sim \mathcal{N} \left( \lambda_{\text{reg}} A^{-1} r^*, \frac{\tau^2}{n} A^{-1} M\Sigma M^\top A^{-\top} \right) , \qquad (23)$$

where the final distribution follows from our assumption on the noise variables $\epsilon_i \sim \mathcal{N}(0, \tau^2)$. Using this above distributional form, we have

$$\mathbb{E}[\|M^*(r^* - \hat{r})\|_{\mathbb{H}_\pi}^2] = \lambda_{\text{reg}}^2 \cdot \langle M^* A^{-1} r^*, M^* A^{-1} r^* \rangle_{\mathbb{H}_\pi} + \frac{\tau^2}{n} \cdot \text{tr}\left[ S_\pi M^* A^{-1} M\Sigma_n M^\top A^{-\top} (M^*)^\top \right]$$

$$= \lambda_{\text{reg}}^2 \cdot \text{tr}\left[ (r^*)^\top A^{-\top} (M^*)^\top S_\pi M^* A^{-1} r^* \right] + \frac{\tau^2}{n} \cdot \text{tr}\left[ S_\pi M^* A^{-1} M\Sigma M^\top A^{-\top} (M^*)^\top \right] . \qquad (24)$$

The final bound for the general policy set $C_\pi$ follows from using the above bound with a an application of Jensen's inequality. In order to convert the above bound to a high probability bound, we require an infinite dimensional analog of the Hanson-Wright concentration inequality. Using Theorem 2.6 from Chen & Yang (2021) along with equation (23), we obtain

$$\Pr(\Delta(\hat{\pi}; r^*) \geq \mathbb{E}[\Delta(\hat{\pi}; r^*)] + t) \leq 2 \exp\left( -C \min\left( \frac{t^2}{\|\Gamma\|_{\text{HS}}^2}, \frac{t}{\|\Gamma\|_{\text{op}}} \right) \right)$$

where the covariance matrix $\Gamma = S_\pi^{\frac{1}{2}} M^* A^{-1} M\Sigma M^\top A^{-\top} (M^*)^\top S_\pi^{\frac{1}{2}}$. $\qquad \square$

### B.1.1 Proof of Lemma 1

Let the vector $y = x + \delta_x$ for some difference vector $\delta_x$. Using this, we have

$$\langle x, \frac{x}{\|x\|} - \frac{y}{\|y\|} \rangle = \langle x, \frac{x}{\|x\|} - \frac{x + \delta_x}{\|x + \delta_x\|} \rangle$$

$$= \frac{\|x\|}{\|x + \delta_x\|} \left( \|x + \delta_x\| - \|x\| - \frac{\langle x, \delta_x \rangle}{\|x\|} \right)$$

$$\overset{(i)}{\leq} \frac{\|x\|}{\|x + \delta_x\|} \left( \|x\| + \frac{\langle x, \delta_x \rangle}{\|x\|} + \frac{\|\delta_x^2\|}{2\|x\|} - \|x\| - \frac{\langle x, \delta_x \rangle}{\|x\|} \right)$$

$$= \frac{\delta_x^2}{2\|x + \delta_x\|} , \qquad (25)$$

where (i) follows from using the inequality $\sqrt{a^2 + z} \leq a + \frac{z}{2a}$. This establishes the result. $\qquad \square$

### B.2 Proof of Proposition 1

Let us denote the the map $M = \text{diag}(\nu_j)$ and the covariance matrix $\Sigma = \text{diag}(\sigma_j)$. From the upper bound obtained in Theorem 1, we have,

$$\mathbb{E}[\|M^*(r^* - \hat{r})\|_{\mathbb{H}_\pi}^2] = \lambda_{\text{reg}}^2 \cdot \|M^* A^{-1} r^*\|_{\mathbb{H}_\pi}^2 + \frac{\tau^2}{n^2} \cdot \sum_{i=1}^{n} \|M^* A^{-1} M\pi_i\|_{\mathbb{H}_\pi}^2$$

$$\overset{(i)}{\leq} \lambda_{\text{reg}}^2 \cdot \|S_\pi^{\frac{1}{2}} M^* A^{-1} S_r^{-\frac{1}{2}}\|_{\text{op}}^2 + \frac{\tau^2}{n} \cdot \text{tr}\left[ S_\pi M^* A^{-1} M\Sigma M^\top A^{-\top} (M^*)^\top \right]$$

$$\overset{(ii)}{\leq} \lambda_{\text{reg}}^2 \cdot \sup_{j \geq 1} \left[ \frac{\nu_j^2 \mu_{r,j} \mu_{\pi,j}}{\nu_j^4 \sigma_j^2 + \lambda_{\text{reg}}^2 \mu_{r,j}^2} \right] + \frac{\tau^2}{n} \cdot \sup_{j \geq 1} \left[ \frac{\nu_j^4 \mu_{\pi,j}^2}{\nu_j^4 \sigma_j^2 + \lambda_{\text{reg}}^2 \mu_{r,j}^2} \right] , \qquad (26)$$

where inequality (i) follows from using the fact that $\|r^*\|_{\mathbb{H}_r} \leq 1$ and inequality (ii) uses the diagonal structure of the map $M$ as well as the fact that each policy $\pi_i \in \mathcal{Q}$ has unit $\mathbb{H}_\pi$-norm.

Recall that the choice of querying strategy queries each scaled eigenfunction $\sqrt{\mu_{\pi,j}}\phi_{\pi,j}$ of the policy space $n^{1-\alpha}$ times. Therefore the $j^{th}$ entry of the covariance matrix $\Sigma$ is given by

$$\sigma_j = \begin{cases} \frac{\mu_{\pi,j}}{n^{\alpha}} & \text{for } j \leq n^{\alpha} \\ 0 & \text{otherwise} \end{cases}. \tag{27}$$

Plugging the above value of $\sigma_j$ into equation (26), we obtain

$$\mathbb{E}[\|M^*(r^* - \hat{r})\|_{\mathbb{H}_\pi}^2] \leq \max\left\{ \sup_{j \leq n^{\alpha}} \frac{\lambda_{\mathsf{reg}}^2 n^{2\alpha} \zeta_j}{\zeta_j^2 + \lambda_{\mathsf{reg}}^2 n^{2\alpha}}, \sup_{j > n^{\alpha}} \zeta_j \right\}$$
$$+ \frac{\tau^2}{n} \cdot \max\left\{ \sup_{j \leq n^{\alpha}} \frac{n^{2\alpha} \zeta_j^2}{\zeta_j^2 + \lambda_{\mathsf{reg}}^2 n^{2\alpha}}, \sup_{j > n^{\alpha}} \frac{\zeta_j^2}{\lambda_{\mathsf{reg}}^2} \right\} \tag{28}$$

This concludes the proof of the proposition. $\square$

### B.3 PROOF OF COROLLARY 1

We now derive explicit finial sample rates for the case when the spectrum of the map $M^\top S_r M S_\pi^{-1}$ satisfies a power law decay for some parameter $\beta > 0$. In the notation used in Proposition 1, we have the quantity

$$\zeta_j \asymp j^{-\beta}. \tag{29}$$

Our proof strategy will be to instantiate the bias and variance terms for this setting of $\zeta_j$ and finally select a setting for the exploration parameter $\alpha$ and regularization parameter $\lambda_{\mathsf{reg}}$.

**Bounding Bias.** The bias term in the proposition is a max over two terms

$$\text{Bias}^2 = \max\left\{ \sup_{j \leq n^{\alpha}} \frac{\lambda_{\mathsf{reg}}^2 n^{2\alpha} \zeta_j}{\zeta_j^2 + \lambda_{\mathsf{reg}}^2 n^{2\alpha}}, \sup_{j > n^{\alpha}} \zeta_j \right\}. \tag{30}$$

We consider the two terms in the analysis here separately. For the first term,

$$\sup_{j \leq n^{\alpha}} \frac{\lambda_{\mathsf{reg}}^2 n^{2\alpha} \zeta_j}{\zeta_j^2 + \lambda_{\mathsf{reg}}^2 n^{2\alpha}} = \lambda_{\mathsf{reg}}^2 \sup_{j \leq n^{\alpha}} \left[ \frac{1}{\frac{j^{-\beta}}{n^{2\alpha}} + \lambda_{\mathsf{reg}}^2 j^{\beta}} \right] \leq \lambda_{\mathsf{reg}} n^{\alpha}, \tag{31}$$

where the final inequality follows from using $a^2 + b^2 \geq 2ab$. For the second term, we have

$$\sup_{j \geq n^{\alpha}} \zeta_j = \sup_{j \geq n^{\alpha}} j^{-\beta} = n^{-\alpha\beta}. \tag{32}$$

**Bounding Variance.** Recall that the variance term (assuming $\tau = 1$) is given by

$$\text{Variance} = \frac{1}{n} \cdot \max\left\{ \sup_{j \leq n^{\alpha}} \frac{n^{2\alpha} \zeta_j^2}{\zeta_j^2 + \lambda_{\mathsf{reg}}^2 n^{2\alpha}}, \sup_{j > n^{\alpha}} \frac{\zeta_j^2}{\lambda_{\mathsf{reg}}^2} \right\}. \tag{33}$$

We again consider both terms of the maximum separately. For the first term,

$$\frac{1}{n} \cdot \sup_{j \leq n^{\alpha}} \frac{n^{2\alpha} \zeta_j^2}{\zeta_j^2 + \lambda_{\mathsf{reg}}^2 n^{2\alpha}} \leq n^{2\alpha - 1}, \tag{34}$$

where the inequality follows from ignoring the term $\lambda_{\mathsf{reg}}^2 n^{2\alpha}$ in the denominator. For the second variance term,

$$\sup_{j > n^{\alpha}} \frac{\zeta_j^2}{n\lambda_{\mathsf{reg}}^2} = \frac{n^{-2\alpha\beta}}{\lambda_{\mathsf{reg}}^2 n}. \tag{35}$$

**Setting regularization parameter.** By setting $\lambda_{\mathsf{reg}} > n^{-\alpha\beta - \alpha}$, we can have that the bias term is dominated by $\lambda_{\mathsf{reg}} n^{\alpha}$. Similarly, the above setting also implies that the variance term is dominated by $n^{2\alpha - 1}$. Combing these observations, we have that the expected error is upper bounded by

$$\Delta(\hat{\pi}_{\mathsf{plug}}; r^*) \leq \lambda_{\mathsf{reg}} n^{\alpha} + n^{2\alpha - 1} \quad \text{where } \lambda_{\mathsf{reg}} > n^{-\alpha\beta - \alpha}. \tag{36}$$

Setting $\lambda_{\mathsf{reg}} = n^{-\alpha(\beta + 1)}$ and then $\alpha = \frac{1}{\beta + 2}$, we get that

$$\Delta(\hat{\pi}_{\mathsf{plug}}; r^*) \leq n^{-\frac{\beta}{\beta + 2}}. \tag{37}$$

This completes the proof of the corollary. $\square$

### B.4  Proof of Theorem 2

In order to prove the general theorem, we exhibit a transformation which allows us to reduce the problem to that with the diagonal structure described in Proposition 1.

We will consider orthogonally diagonalizable matrices $S_r$ and $S_\pi$ which represent the eigenvectors and eigenvalues of the Hilbert spaces $\mathbb{H}_r$ and $\mathbb{H}_\pi$. Consider the following set of transformations for any reward $r \in C_r$ and policy $\pi \in C_\pi$.

$$\tilde{r} = S_r^{\frac{1}{2}} r, \quad \tilde{\pi} = S_\pi^{\frac{1}{2}} \pi, \quad \tilde{M} = S_r^{\frac{1}{2}} M S_\pi^{-\frac{1}{2}}. \tag{38}$$

With this transformation, we can rewrite the objective function above

$$\max_{\tilde{\pi}} \langle \tilde{r}, \tilde{M}\tilde{\pi} \rangle \quad \text{s.t.} \quad \langle \tilde{\pi}, \tilde{\pi} \rangle = 1 \text{ and } \langle \tilde{r}, \tilde{r} \rangle = 1 \, ,$$

where the inner product $\langle \cdot, \cdot \rangle$ denotes the standard $\ell_2$ inner product. Observe that we have overloaded notation to denote by $\tilde{r}^* = \tilde{r}$. Further, using these above transformations, we can rewrite the adjoint operator

$$M^* = S_\pi^{-1} M^\top S_r = S_\pi^{-\frac{1}{2}} (S_r^{\frac{1}{2}} M S_\pi^{-\frac{1}{2}})^\top S_r^{\frac{1}{2}} = S_\pi^{-\frac{1}{2}} \tilde{M}^\top S_r^{\frac{1}{2}} \, . \tag{39}$$

Recall from Theorem 1, the matrix

$$A = M\Sigma M^\top S_r + \lambda_{\text{reg}} I = S_r^{-\frac{1}{2}} \left[ \tilde{M}\tilde{\Sigma}\tilde{M}^\top + \lambda_{\text{reg}} I \right] S_r^{\frac{1}{2}} \, , \tag{40}$$

where the covariance matrix $\tilde{\Sigma} = \frac{1}{n} \sum_i \tilde{\pi}\tilde{\pi}^\top$. We have used the fact here that the matrices $S_\pi$ and $S_r$ are orthogonally diagonalizable and hence symmetric. Finally, we will denote the singular value decomposition of the compact map $M$ in the matrix form as

$$\tilde{M} = U_{\tilde{M}} \Lambda_{\tilde{M}} V_{\tilde{M}}^\top \, .$$

The existence of such a decomposition is guaranteed by the regularity assumptions we consider on the map $M$ in Appendix A. We will now analyze the bias and the variance terms from the upper bound on $\mathbb{E}[\|M^*(r^* - \hat{r}\|_{\mathbb{H}_\pi}^2]$ from Theorem 1.

**Bound on bias.**  The squared bias term is given by

$$
\begin{aligned}
\lambda_{\text{reg}}^{-2} \cdot \text{Bias}^2 &= r^\top A^{-\top} (M^*)^\top S_\pi M^* A^{-1} r \\
&= r^\top S_r^{\frac{1}{2}} S_r^{-\frac{1}{2}} \cdot S_r^{\frac{1}{2}} (\tilde{M}\tilde{\Sigma}\tilde{M}^\top + \lambda_{\text{reg}} I)^{-1} S_r^{-\frac{1}{2}} \cdot S_r^{\frac{1}{2}} \tilde{M} S_\pi^{-\frac{1}{2}} \cdot S_\pi \cdot M^* A^{-1} r \\
&= \tilde{r}^\top (\tilde{M}\tilde{\Sigma}\tilde{M}^\top + \lambda_{\text{reg}} I)^{-1} \tilde{M} \cdot S_\pi^{\frac{1}{2}} S_\pi^{-\frac{1}{2}} \tilde{M}^\top S_r^{\frac{1}{2}} \cdot S_r^{-\frac{1}{2}} (\tilde{M}\tilde{\Sigma}\tilde{M}^\top + \lambda_{\text{reg}} I)^{-1} S_r^{\frac{1}{2}} r \\
&= \tilde{r}^\top (\tilde{M}\tilde{\Sigma}\tilde{M}^\top + \lambda_{\text{reg}} I)^{-1} \tilde{M} \cdot \tilde{M}^\top (\tilde{M}\tilde{\Sigma}\tilde{M}^\top + \lambda_{\text{reg}} I)^{-1} \tilde{r} \\
&= \tilde{r}^\top U_{\tilde{M}} (\Lambda_{\tilde{M}} V_{\tilde{M}}^\top \tilde{\Sigma} V_{\tilde{M}} \Lambda_{\tilde{M}} + \lambda_{\text{reg}} I)^{-1} \Lambda_{\tilde{M}}^2 (\Lambda_{\tilde{M}} V_{\tilde{M}}^\top \tilde{\Sigma} V_{\tilde{M}} \Lambda_{\tilde{M}} + \lambda_{\text{reg}} I)^{-1} U_{\tilde{M}}^\top \tilde{r} \, , \tag{41}
\end{aligned}
$$

where we have used the SVD decomposition for the matrix $\tilde{M}$ in the last step.

**Bound on variance.**  The variance term is given by

$$
\begin{aligned}
\text{Var} &= \frac{\tau^2}{n} \cdot \text{tr} \left[ S_\pi M^* A^{-1} M \Sigma_n M^\top A^{-\top} (M^*)^\top \right] \\
&= \frac{\tau^2}{n} \cdot \text{tr} \left[ \tilde{M}^\top (\tilde{M}\tilde{\Sigma}\tilde{M}^\top + \lambda_{\text{reg}} I)^{-1} \tilde{M}\tilde{\Sigma}\tilde{M}^\top (\tilde{M}\tilde{\Sigma}\tilde{M}^\top + \lambda_{\text{reg}} I)^{-1} \tilde{M} \right] \\
&= \frac{\tau^2}{n} \cdot \text{tr} \left[ \Lambda_{\tilde{M}} (\Lambda_{\tilde{M}} V_{\tilde{M}}^\top \tilde{\Sigma} V_{\tilde{M}} \Lambda_{\tilde{M}} + \lambda_{\text{reg}} I)^{-1} \Lambda_{\tilde{M}} V_{\tilde{M}}^\top \tilde{\Sigma} V_{\tilde{M}} \Lambda_{\tilde{M}} (\Lambda_{\tilde{M}} V_{\tilde{M}}^\top \tilde{\Sigma} V_{\tilde{M}} \Lambda_{\tilde{M}} + \lambda_{\text{reg}} I)^{-1} \Lambda_{\tilde{M}} \right] \, . \tag{42}
\end{aligned}
$$

Finally, by making a substitution for reward $\tilde{r} = U_{\tilde{M}}^\top \tilde{r}$ and policy $\tilde{\pi} = V_{\tilde{M}}^\top \tilde{\pi}$ in equations (41) and (42), we recover back the bias variance expressions used in the analysis for Proposition 1. What

remains to be shown is that our particular choice of query policies correspond to basis vectors in this transformed space. For this, observe that the sampling policies

$$\pi_j = \sum_{i=1}^{\infty} \sqrt{\mu_{\pi,i}} \cdot \langle \phi_{\tilde{M},j}, \phi_{\pi,i} \rangle \phi_{\pi,i} \quad \text{for } j \leq n^{\alpha} \,,$$

is such that the transformed policies

$$\tilde{\pi}_j = V_{\tilde{M}}^{\top} S_{\pi}^{\frac{1}{2}} \pi_j = V_{\tilde{M}}^{\top} S_{\pi}^{\frac{1}{2}} \cdot S_{\pi}^{-\frac{1}{2}} V_{\tilde{M}} e_j = e_j \,, \tag{43}$$

indeed correspond to the basis vector. This finishes the proof of the desired claim. $\qquad \square$

### B.5 PROOF OF COROLLARY 2

The proof of this corollary follows similar to that of Corollary 1 in terms of bounding the bias and the variance. The final rate follows by an application of Jensen's inequality to conclude

$$\mathbb{E}[\|M^*(r^* - \hat{r})\|_{\mathbb{H}_\pi}] \leq (\mathbb{E}[\|M^*(r^* - \hat{r})\|_{\mathbb{H}_\pi}^2])^{\frac{1}{2}} \,. \tag{44}$$

The final rate that we get in this case is thus upper bounded by the square root of the rate observed in Corollary 1. This concludes the proof. $\qquad \square$

## C GAUSSIAN PROCESS BANDIT OPTIMIZATION

In this section, we discuss in detail the application of our framework to the problem of frequentist Gaussian process bandit optimization, also known as Kernelized multi-armed bandits (MAB) problem. Recall the reduction of the Kernel MAB problem to our setup required us to define three elements.

**Reward space $\mathbb{H}_r$.** Given the RKHS $\mathbb{H}$ as well as the elements of the cover $\mathcal{C}_\epsilon$, we view the reward function as a map from $\mathcal{C}_\epsilon$ to $\mathbb{R}$, or equivalently as a vector in $\mathbb{R}^{N_{\text{cov}}(\epsilon)}$. More precisely, letting $\tilde{f} = [f(x_1), \ldots, f(x_{N_{\text{cov}}})]$ denote the vector of evaluations of a function $f$, we define

$$\begin{aligned} \mathbb{H}_r &:= \text{span}\{\tilde{f} \mid f \in \mathbb{H}\} \\ \text{with } \langle \tilde{f}_1, \tilde{f}_2 \rangle_{\mathbb{H}_r} &:= \tilde{f}_1^{\top} K^{-1} \tilde{f}_2 \end{aligned} \tag{45}$$

where $\langle \cdot, \cdot \rangle$ represents the standard $\ell_2$ inner product. With this notation, let us define the true reward $r^* := \tilde{f}^* = [f^*(x_1), \ldots, f^*(x_{N_{\text{cov}}})]$.

**Policy Space $\mathbb{H}_\pi$.** For the policy space $\mathbb{H}_\pi$ in our setup, we let

$$\begin{aligned} \mathbb{H}_\pi &:= \text{span}\{k_x = [\mathcal{K}(x, x_1), \ldots, \mathcal{K}(x, x_{N_{\text{cov}}})] \in \mathbb{R}^{N_{\text{cov}}} \mid x \in \mathcal{C}_\epsilon\} \\ \text{with } \langle k_1, k_2 \rangle_{\mathbb{H}_\pi} &:= \langle k_1, K^{-2} k_2 \rangle \,. \end{aligned} \tag{46}$$

The choice of the above norm ensures that

$$\langle k_i, k_j \rangle_{\mathbb{H}_\pi} = \langle K^{-1} k_i, K^{-1} k_j \rangle = \langle e_i, e_j \rangle = \delta_{i,j} \quad \text{for all} \quad (x_i, x_j) \in \mathcal{C}_\epsilon \times \mathcal{C}_\epsilon \,.$$

For the policy space $\mathbb{H}_\pi$, we have created an orthonormal embedding of the set of vectors $\{k_x\}_{x \in \mathcal{C}}$. Observe that this policy set that we construct satisfies the regularity Assumption 1 because each vector $k$ is an eigenvector of the space $\mathbb{H}_\pi$.

**Map $M$.** By our assumption that the kernel $\mathcal{K}$ is a Mercer's kernel, we have that $\mathbb{H}_\pi \subseteq \mathbb{H}_r$, that is, for all $x \in \mathcal{C}$, the vector $k_x \in \mathbb{H}_r$. Furthermore, both $\mathbb{H}_r$ and $\mathbb{H}_\pi$ are sub-spaces of $\mathbb{R}^{N_{\text{cov}}}$ and we can take the map $M = I_{N_{\text{cov}}}$.

With these definitions, we now explicitly establish a correspondence between our doubly nonparameteric bandit problem and the Kernel MAB problem.

## C.1 CONNECTING THE PROBLEMS

Given an RKHS $\mathbb{H}$ with an associated Mercer's kernel $\mathcal{K}$, the objective of the zeroth-order bandit optimization problem is

$$\max_{x \in \mathcal{X}} f^*(x) \quad \text{such that} \quad \|f^*\|_{\mathbb{H}} \leq 1 , \tag{P1}$$

with access to oracle

$$\mathcal{O}_{f^*} : x \mapsto f^*(x) + \eta \quad \text{where } \eta \sim \mathcal{N}(0, \tau^2) .$$

Equivalently, the objective in our reward learning framework is

$$\max_{\pi \in \mathbb{H}_{\pi}} \langle r^*, \pi \rangle_{\mathbb{H}_r} \quad \text{such that} \quad \|r^*\|_{\mathbb{H}_r} \leq 1 \text{ and } \|\pi\|_{\mathbb{H}_{\pi}} \leq 1 , \tag{P2}$$

with the corresponding spaces and inner products are defined in the previous section. The oracle required in our setup responds with

$$\mathcal{O}_{r^*} : \pi \mapsto \langle r^*, \pi \rangle_{\mathbb{H}_r} + \eta \quad \text{where } \eta \sim \mathcal{N}(0, \tau^2) ,$$

for any policy $\pi \in \mathbb{H}_{\pi}$ such that $\|\pi\|_{\mathbb{H}_{\pi}} \leq 1$. Our first lemma below states that obtaining such a n oracle is indeed feasible if we are able to restrict our queries $\pi$ to include only points $k_x$ for which the vector $k_x \in \mathcal{C}_{\epsilon}$.

**Lemma 2.** *Given access to oracle $\mathcal{O}_{f^*}$ for a function $f^*$, the corresponding oracle $\mathcal{O}_{r^*}$ can be implemented when the query set consists of $\{k_x\}_{x \in \mathcal{C}_{\epsilon}}$.*

*Proof.* For any query point $k$, the oracle $\mathcal{O}_{r^*}$ needs to compute the value $\langle r^*, k \rangle_{\mathbb{H}_r} = f^*(x)$. Thus, these two oracles on the provided query set are exactly identical. $\square$

**Lemma 3.** *For any $f^* \in \mathbb{H}$ satisfying $\|f^*\|_{\mathbb{H}} \leq 1$, we have that $\|r^*\|_{\mathbb{H}_r} \leq 1$.*

*Proof.* Observe that an alternate way to define the RKHS norm is given by

$$\|f\|_{\mathbb{H}} := \sup_{S \subseteq \mathcal{X}; |S| \leq \infty} f|_S K_S^{-1} f|_S .$$

The fact that $\|r^*\|_{\mathbb{H}_r}$ is computed on $\mathcal{C}_{\epsilon} \subset \mathcal{X}$ establishes the desired claim. $\square$

Finally, we turn to establishing a relation between the solutions obtained from solving the relaxed problem (P2) as compared to solving the original problem (P1). We denote the corresponding maximizers for both problems

$$x^* \in \arg\max_{x \in \mathcal{X}} f^*(x) \quad \text{and} \quad x_{\pi}^* \in \arg\max_{x \in \mathcal{C}_{\epsilon}} \langle r^*, k_x \rangle_{\mathbb{H}_r} , \tag{47}$$

The following lemma now relates both these maximizers together.

**Lemma 4.** *For an RKHS $\mathbb{H}$ with kernel $\mathcal{K}$ satisfying Assumption 2 with constant $L_{\mathcal{K}} > 0$ and any function $f^* \in \mathbb{H}$, let $x^* \in \mathcal{X}$ and $x_{\pi}^* \in \mathcal{C}_{\epsilon}$ be the maximizers as defined in equation (47), we have*

$$f^*(x_{\pi}^*) \geq f^*(x^*) - \sqrt{2cL_{\mathcal{K}}\epsilon} . \tag{48}$$

*Proof.* Denote by $\Pi_{\mathcal{C}_{\epsilon}}(x^*) := \arg\min_{x \in \mathcal{C}_{\epsilon}} \|x^* - x\|_2$ the projection of the point $x^*$ onto the set $\mathcal{C}_{\epsilon}$. Then, we have

$$f^*(x^*) - f^*(x_{\pi}^*) = f^*(x^*) - f^*(\Pi_{\mathcal{C}_{\epsilon}}(x^*)) + f^*(\Pi_{\mathcal{C}_{\epsilon}}(x^*)) - f^*(x_{\pi}^*)$$
$$\leq \sqrt{2cL_{\mathcal{K}}\epsilon} .$$

This completes the proof of the lemma. $\square$

The above lemma shows that solving Problem P2 is equivalent to solving Problem P1 up to an additive factor of $\sqrt{2cL_{\mathcal{K}}\epsilon}$ when we are working with an $\epsilon$-cover over the domain space.

## C.2 ANALYSIS FOR BANDIT OPTIMIZATION

Recall from the previous section that the quantity which determines the rate of decay is the ratio of eigenvalues

$$\zeta_j = \frac{\hat{\mu}_{\pi,j}}{\hat{\mu}_{r,j}} = \frac{\hat{\mu}_{r,j}^2}{\hat{\mu}_{r,j}} = \hat{\mu}_{r,j} \ ,$$

where $\hat{\mu}_{r,j}$ is the $j^{th}$ eigenvalue of the kernel matrix $K$. Let us denote by $\mathbb{P}$ denote the uniform distribution over the input space $\mathcal{X}$ and let us suppose that the cover $N_{\text{cov}}$ is formed using random samples from this distribution. Let us denote by $\{\mu_j\}$ the eigenvalues and by $\phi_j$ the corresponding eigen vectors of the Mercer kernel $\mathcal{K}$. For every point $x \in \mathcal{X}$, let us denote by

$$\Phi(x) := \left( \sqrt{\mu_j} \phi_j(x) \right)_{j=1}^{\infty} \ ,$$

the corresponding featurization of the point $x$. Then, for $S := \mathbb{E}_{x \sim \mathbb{P}}[\Phi(x)\Phi(x)^{\top}]$, we have

$$[S]_{j,k} = [\mathbb{E}_{x \sim \mathbb{P}}[\Phi(x)\Phi(x)^{\top}]]_{j,k} = \mathbb{E}_{x \sim \mathbb{P}}[\sqrt{\mu_j}\sqrt{\mu_k}\phi_j(x)\phi_k(x)] = \mu_j \delta_{j,k} \ . \tag{49}$$

Observe that the kernel matrix $K$ and the (scaled) sample covariance matrix $N_{\text{cov}} \cdot \hat{S} = \sum_{x \in \mathcal{C}} \Phi(x)\Phi(x)^{\top}$ are similar matrices and thus have the same eigenvalues. The following lemma, adapted from Koltchinskii & Lounici (2017, Theorem 9) relates the eigenvalues of the sample covariance matrix $\hat{S}$ to those of the underlying kernel $\mathcal{K}$.

**Lemma 5.** *For any $\lambda_S > 0$ and size of the cover satisfying $N_{\text{cov}}(\epsilon) > c \cdot \frac{\text{tr}(S(S+\lambda_S I)^{-1})}{\epsilon_S^2} + \frac{1}{\epsilon_S^2} \log\left(\frac{1}{\delta}\right)$, we have,*

$$\hat{\mu}_j \leq (1 + \epsilon_S)\mu_j + \lambda_S \epsilon_S \quad \text{for all } j \ , \tag{50}$$

*with probability at least $1 - \delta$.*

The following corollary of Lemma 5 provides us with a way to control the deviation of the eigenvalues $\hat{\mu}_j$ from the corresponding $\mu_j$ in a multiplicative manner.

**Corollary 4.** *For any value of decay parameter $\beta > 1$ and $\gamma < \beta$, we have, for all $j$, the eigenvalues*

$$\hat{\mu}_j \leq \frac{3}{2}\mu_j + \frac{N_{\text{cov}}^{-\gamma}}{2} \ , \tag{51}$$

*with high probability.*

*Proof.* Let us understand the condition $N_{\text{cov}}(\epsilon) \gg \frac{\text{tr}(S(S+\lambda_S I)^{-1})}{\epsilon_S^2}$ and see what restrictions it puts on the value of the covering number. Lets suppose that the true eigen values $\mu_j \asymp j^{-\beta}$ and we set the value of $\lambda_S \asymp N_{\text{cov}}^{-\gamma}$. Therefore, the sum

$$\sum_j \frac{j^{-\beta}}{j^{-\beta} + \lambda_S} \lesssim N_{\text{cov}}^{\frac{\gamma}{\beta}} + \frac{1}{N_{\text{cov}}^{-\gamma}} \sum_{j > N_{\text{cov}}^{\frac{\gamma}{\beta}}} j^{-\beta}$$

$$\lesssim N_{\text{cov}}^{\frac{\gamma}{\beta}} + \frac{N_{\text{cov}}^{\frac{\gamma}{\beta}}}{\beta - 1} \ .$$

Thus, if we set $\epsilon_S = \frac{1}{2}$, then for any $\beta > 1$ and $\gamma < \beta$, the above condition on the covering number will be satisfied and we get desired bound on the deviation of the empirical eigenvalues from population eigenvalues. $\square$

The above corollary is essential to our argument because often times we have a good understanding of the decay of the eigenvalues of the kernel $\mathcal{K}$ associated with the RKHS and this allows us to relate the set of empirical eigenvalues to these.

We now present a proof of Theorem 3, restated below, which upper bounds the excess risk for this setup. We will then use a batch to online conversion bound to convert this to a regret bound and specialize to the Matérn kernel later.

**Theorem 5** (Restated Theorem 3). *Suppose that the eigenvalues of a $L_{\mathcal{K}}$-Lipschitz kernel $\mathcal{K}$ with respect to a distribution $\mathbb{P}$ over $\mathcal{X}$ satisfy the power-law decay $\mu_j \asymp j^{-\beta}$. Let $\hat{x}_{\mathsf{plug}}$ be the output of Algorithm 1 using $n$ queries to the oracle $\mathcal{O}_{f^*}$. Then, for any value of $\gamma \in (1 + \frac{1}{d} \frac{\log(1/\epsilon)}{\log(L_{\mathcal{K}}/\epsilon^2)}, \beta)$ and $\epsilon \in (0,1)$, the excess risk*

$$\max_x f^*(x) - f^*(\hat{x}_{\mathsf{plug}}) \lesssim N_{\mathsf{cov}}^{\frac{1}{\beta+2}}(\epsilon) \cdot n^{\frac{-\beta}{2(\beta+2)}} + N_{\mathsf{cov}}^{\frac{1-\gamma}{2}}(\epsilon) + \sqrt{L_{\mathcal{K}}\epsilon} \,,$$

*with high probability.*

*Proof.* Our strategy, as before, will be to explore $n^{\alpha}$ directions and assume $\tau^2 = 1$. Recall, that for symmetric matrices, Theorem 2, the excess error of the plug-in estimator can be upper bounded as

$$\mathbb{E}[\Delta(\hat{\pi}_{\mathsf{plug}}; r^*)]^2 \leq \lambda_{\mathsf{reg}}^2 \sup_{j \geq 1} \left[ \frac{1}{\frac{\nu_j^2 \sigma_j^2}{\mu_{\pi,j}\mu_{r,j}} + \frac{\lambda_{\mathsf{reg}}^2 \mu_{r,j}}{\mu_{\pi,j}\nu_j^2}} \right] + \frac{1}{n} \sup_{j \geq 1} \left[ \frac{\nu_j^4 \mu_{\pi,j}^2}{\nu_j^4 \sigma_j^2 + \lambda_{\mathsf{reg}}^2 \mu_{r,j}^2} \right] \,.$$

**Bounding Bias.** We will split the analysis into two cases.

Case 1: $j > n^{\alpha}$. For this case, we have that $\sigma_j = 0$ and therefore

$$\lambda_{\mathsf{reg}}^2 \sup_{j > n^{\alpha}} \frac{\hat{\mu}_{\pi,j}}{\lambda_{\mathsf{reg}}^2 \hat{\mu}_{r,j}} = \sup_{j > n^{\alpha}} N_{\mathsf{cov}} \hat{\mu}_j \lesssim \sup_{j > n^{\alpha}} N_{\mathsf{cov}}(\mu_j + N_{\mathsf{cov}}^{-\gamma}) \leq N_{\mathsf{cov}} n^{-\alpha\beta} + N_{\mathsf{cov}}^{1-\gamma} \,, \tag{52}$$

with the above holding with high probability from an application of Corollary 4 for any $1 < \gamma < \beta$.

Case 2: $j \leq n^{\alpha}$. For this case, we have $\sigma_j = \frac{\mu_{\pi,j}}{n^{\alpha}}$. The bias can then be upper bounded as

$$\lambda_{\mathsf{reg}}^2 \sup_{j \leq n^{\alpha}} \left[ \frac{1}{\frac{\nu_j^2 \mu_{\pi,j}}{n^{2\alpha}\mu_{r,j}} + \frac{\lambda_{\mathsf{reg}}^2 \mu_{r,j}}{\mu_{\pi,j}\nu_j^2}} \right] \leq \lambda_{\mathsf{reg}} n^{\alpha} \,, \tag{53}$$

where the final inequality follows from using $a^2 + b^2 \geq 2ab$.

**Bounding variance.** As we did in the section above, let us split the analysis into two cases.

Case 1: $j > n^{\alpha}$. For this case, the variance term simplifies to

$$\frac{1}{n} \sup_{j > n^{\alpha}} \left[ \frac{\mu_{\pi,j}^2}{\lambda_{\mathsf{reg}}^2 \mu_{r,j}^2} \right] = \frac{1}{\lambda_{\mathsf{reg}}^2 n} \sup_{j > n^{\alpha}} \left[ N_{\mathsf{cov}}^2 \hat{\mu}_j^2 \right] \leq \frac{N_{\mathsf{cov}}^2}{\lambda_{\mathsf{reg}}^2 n} \sup_{j > n^{\alpha}} \left[ \hat{\mu}_j^2 \right] \lesssim \frac{N_{\mathsf{cov}}^2 n^{-2\alpha\beta} + N_{\mathsf{cov}}^{2(1-\gamma)}}{\lambda_{\mathsf{reg}}^2 n} \,. \tag{54}$$

Case 2: $j \leq n^{\alpha}$. For the second case, we can upper bound the variance term

$$\frac{1}{n} \sup_{j \leq n^{\alpha}} \left[ \frac{\nu_j^4 \mu_{\pi,j}^2}{\frac{\nu_j^4 \mu_{\pi,j}^2}{n^{2\alpha}} + \lambda_{\mathsf{reg}}^2 \mu_{r,j}^2} \right] \leq \frac{n^{2\alpha}}{n} \,, \tag{55}$$

where the last inequality follows from ignoring the second term in the denominator.

**Setting regularization parameter.** From the analysis in the above paragraphs, we have

$$\mathsf{Bias}^2 \leq \max\{N_{\mathsf{cov}} n^{-\alpha\beta} + N_{\mathsf{cov}}^{1-\gamma}, \lambda_{\mathsf{reg}} n^{\alpha}\} \leq \max\{N_{\mathsf{cov}} n^{-\alpha\beta}, \lambda_{\mathsf{reg}} n^{\alpha}\} + N_{\mathsf{cov}}^{1-\gamma} \,, \tag{56}$$

$$\mathsf{Variance} \leq \max\{\frac{N_{\mathsf{cov}}^2 n^{-2\alpha\beta} + N_{\mathsf{cov}}^{2(1-\gamma)}}{\lambda_{\mathsf{reg}}^2 n}, \frac{n^{2\alpha}}{n}\} \leq \max\{\frac{N_{\mathsf{cov}}^2 n^{-2\alpha\beta}}{\lambda_{\mathsf{reg}}^2 n}, \frac{n^{2\alpha}}{n}\} + \frac{N_{\mathsf{cov}}^{2(1-\gamma)}}{\lambda_{\mathsf{reg}}^2 n} \,. \tag{57}$$

For regularization parameter $\lambda_{\mathsf{reg}} > N_{\mathsf{cov}} n^{-\alpha\beta-\alpha}$ and $\gamma > \frac{\alpha\beta}{\log_n N_{\mathsf{cov}}}$, we have

$$\mathsf{Bias}^2 \leq \lambda_{\mathsf{reg}} n^{\alpha} + N_{\mathsf{cov}}^{1-\gamma} \,,$$

$$\mathsf{Variance} \leq \frac{n^{2\alpha}}{n} \,.$$

**Excess risk bound.** To obtain the final excess risk bound, we set $\alpha = \frac{1 + \log_n N_{\text{cov}}}{\beta + 2}$

$$
\mathbb{E}[\Delta(\hat{\pi}_{\text{plug}}; r^*)]^2 \leq \lambda_{\text{reg}} n^\alpha + \frac{n^{2\alpha}}{n} + N_{\text{cov}}^{1-\gamma}
$$

$$
\leq N_{\text{cov}} n^{-\alpha\beta} + n^{2\alpha - 1} + N_{\text{cov}}^{1-\gamma}
$$

$$
\overset{(i)}{\lesssim} N_{\text{cov}}^{\frac{2}{\beta+2}} n^{\frac{-\beta}{\beta+2}} + N_{\text{cov}}^{1-\gamma} , \tag{58}
$$

where inequality (i) follows from our particular choice of $\alpha$. Combining the above bound with Lemma 4 completes the proof. $\qquad\square$

The following corollary instantiates the above theorem for the case when the input space is the unit ball, that is, $\mathcal{X} = \mathbb{B}_d(1)$.

**Corollary 5.** *Let the input space $\mathcal{X} = \mathbb{B}_d(1)$ and the kernel $\mathcal{K}$ satisfy Assumption 2. Then, for any $\beta > 1 + \frac{2}{d}$, we have*

$$
\max_x f^*(x) - \mathbb{E}_{x \sim \hat{\pi}_{\text{plug}}} f^*(x) \lesssim L_{\mathcal{K}}^{\frac{d}{\beta+2+2d}} n^{\frac{-\beta}{2(\beta+2+2d)}} . \tag{59}
$$

*Proof.* From the bound in Theorem 3, we have,

$$
\max_x f^*(x) - \mathbb{E}_{x \sim \hat{\pi}_{\text{plug}}} f^*(x) \lesssim N_{\text{cov}}^{\frac{1}{\beta+2}}(\epsilon) \cdot n^{\frac{-\beta}{2(\beta+2)}} + N_{\text{cov}}^{\frac{1-\gamma}{2}}(\epsilon) + \sqrt{L_{\mathcal{K}}\epsilon}
$$

$$
\overset{(i)}{\leq} N_{\text{cov}}^{\frac{1}{\beta+2}}\left(\frac{\epsilon^2}{L_{\mathcal{K}}}\right) \cdot n^{\frac{-\beta}{2(\beta+2)}} + N_{\text{cov}}^{\frac{1-\gamma}{2}}\left(\frac{\epsilon^2}{L_{\mathcal{K}}}\right) + \epsilon
$$

$$
\overset{(ii)}{\leq} L_{\mathcal{K}}^{\frac{d}{\beta+2}} \cdot \left(\frac{1}{\epsilon}\right)^{\frac{2d}{\beta+2}} \cdot n^{\frac{-\beta}{2(\beta+2)}} + \left(\frac{L_{\mathcal{K}}}{\epsilon^2}\right)^{\frac{d(1-\gamma)}{2}} + \epsilon
$$

$$
\overset{(iii)}{\leq} L_{\mathcal{K}}^{\frac{d}{\beta+2}} \cdot \left(\frac{1}{\epsilon}\right)^{\frac{2d}{\beta+2}} \cdot n^{\frac{-\beta}{2(\beta+2)}} + 2\epsilon , \tag{60}
$$

where inequality (i) follows from substituting $\epsilon \rightarrow \epsilon^2/L_{\mathcal{K}}$, (ii) follows from the fact that $N_{\text{cov}}(\epsilon) \asymp \left(\frac{1}{\epsilon}\right)^d$, and (iii) follows from using the assumption that $\beta > \gamma > 1 + \frac{2}{d} \frac{\log(1/\epsilon)}{\log(L_{\mathcal{K}}/\epsilon^2)}$.

Finally, setting $\epsilon \asymp L_{\mathcal{K}}^{\frac{d}{\beta+2+2d}} n^{\frac{-\beta}{2(\beta+2+2d)}}$, we get

$$
\max_x f^*(x) - \mathbb{E}_{x \sim \hat{\pi}_{\text{plug}}} f^*(x) \lesssim L_{\mathcal{K}}^{\frac{d}{\beta+2+2d}} n^{\frac{-\beta}{2(\beta+2+2d)}} .
$$

This establishes the desired claim. $\qquad\square$

### C.3 REGRET BOUND FOR MATÉRN KERNEL

In this section, we specialize the bound from Theorem 3 for the special class of Matérn kernels. Recall that the Matern kernel is a distanced based kernel with $\mathcal{K}(x, y) = f(\|x - y\|)$. Denote by $r = \|x - y\|$, the exact form for the kernel is given by

$$
\mathcal{K}_{\text{Matern}, \nu}(r) = \frac{2^{1-\nu}}{\Gamma(\nu)} \left(\frac{\sqrt{2\nu} r}{l}\right)^\nu K_\nu \left(\frac{\sqrt{2\nu} r}{l}\right) , \tag{61}
$$

with parameters $\nu$ and $l$ and where $K_\nu$ is the modified Bessel function of the second kind. Going forward, lets fix the scale parameter $l = 1$ without loss of generality.

The following lemma then bounds the Lipschitz constant for this class of kernels when the distance function is the $\ell_2$ norm.

**Lemma 6** (Lipschitz Matérn Kernel). *Consider the Matérn kernel with parameter $\nu > \frac{3}{2}$. The Lipschitz constant of this kernel is bounded by*

$$
L_{\mathcal{K}} \leq \sup_{r \in (0,1)} \left[\frac{e 2^{2-\nu} \nu K_{\nu-1}(1)}{\Gamma(\nu)} \cdot r e^{-\sqrt{2\nu} r}\right]. \tag{62}
$$

*Proof.* The approach will be to show that the kernel function $\mathcal{K}_{\mathsf{Matern},\nu}$ is a Lipschitz function of the distance $r$ and then cover the $\ell_2$ ball in the $d$ dimensional space appropriately. We now look at the derivative of the function $\mathcal{K}_{\mathsf{Matern},\nu}(r)$ with respect to $r$.

$$
\begin{aligned}
\partial \mathcal{K}_{\mathsf{Matern},\nu}(r) &= \frac{2^{1-\nu}(\sqrt{2\nu})^\nu}{\Gamma(v)}\left(\nu r^{\nu-1}K_\nu(\sqrt{2\nu}r)\partial r + r^\nu \partial K_\nu(\sqrt{2\nu}r)\right) \\
&\stackrel{\text{(i)}}{=} \frac{2^{1-\nu}(\sqrt{2\nu})^\nu}{\Gamma(v)}\left(\nu r^{\nu-1}K_\nu(\sqrt{2\nu}r) - r^\nu\left(\sqrt{2\nu}K_{\nu-1}(\sqrt{2\nu}r) + \frac{\nu\sqrt{2\nu}}{\sqrt{2\nu}r}K_\nu(\sqrt{2\nu}r)\right)\right)\partial r \\
&= -\frac{2^{1-\nu}(\sqrt{2\nu})^\nu}{\Gamma(v)}\left(r^\nu\sqrt{2\nu}K_{\nu-1}(\sqrt{2\nu}r)\right)\partial r ,
\end{aligned}
\tag{63}
$$

where (i) follows from the identity $\partial K_\nu(z) = (-K_{\nu-1}(z) - \frac{\nu}{z}K_\nu(z))\partial z$.

For any $\nu > \frac{1}{2}$, we have the inequality

$$
\frac{K_\nu(x)}{K_\nu(y)} < \exp^{y-x}\left(\frac{y}{x}\right)^\nu \quad \text{for } 0 < x < y.
\tag{64}
$$

Instantiating the above with $y = 1$ and $\nu > \frac{3}{2}$, we have

$$
\begin{aligned}
|\partial\mathcal{K}_{\mathsf{Matern},\nu}(r)| &\leq \frac{2^{1-\nu}(\sqrt{2\nu})^\nu}{\Gamma(v)}\left(r^\nu\sqrt{2\nu}\cdot\frac{e^{-\sqrt{2\nu}r}}{(\sqrt{2\nu}r)^{\nu-1}}\cdot eK_{\nu-1}(1)\right) \\
&\leq \frac{e2^{2-\nu}\nu K_{\nu-1}(1)}{\Gamma(\nu)}\cdot re^{-\sqrt{2\nu}r} .
\end{aligned}
\tag{65}
$$

The Lipschitz constant for this case can now be obtained by taking a sup over $r \in (0,1)$. $\qquad\square$

While our upper bound was in terms of sample complexity, in order to compete with the cumulative regret formulation, we adapt an explore-then-commit strategy. The following lemma relates the sample complexity bound to a cumulative regret bound.

**Lemma 7** (Batch to online conversion). *Suppose an algorithm has sample complexity $O(n^{-\alpha})$ in the passive learning setup, the explore then commit strategy based on this learning algorithm would have regret $O(T^{\frac{1}{1+\alpha}})$.*

*Proof.* For some parameter $\gamma > 0$, let the explore then commit algorithm explore for $T^\gamma$ steps and the commit to the strategy obtained post this exploration for the remaining $T - T^\gamma$ time steps. The cumulative regret for such an algorithm is

$$
\mathfrak{R}_T = T^\gamma + T^{-\alpha\gamma}(T - T^\gamma) \leq T^\gamma + T^{1-\alpha\gamma} .
\tag{66}
$$

Setting $\gamma = \frac{1}{1+\alpha}$ finishes the proof. $\qquad\square$

We now proceed to prove Corollary 3 which instantiates the bound in Theorem 3 for the class of Matérn kernels.

**Corollary 6** (Restated Corollary 3). *Consider the family of Matérn kernels with parameter $\nu > \frac{3}{2}$ defined with the euclidean norm over $\mathbb{R}^d$. The $T$-step regret of the explore-then-commit algorithm is*

$$
\mathfrak{R}_{\mathsf{Mat},T} \lesssim O\left(L_{\mathcal{K}}^{\frac{d^2}{2\nu+d(3+2d)}}\cdot T^{\frac{4\nu+d(6+4d)}{6\nu+d(7+4d)}}\right) .
$$

*with high probability.*

*Proof.* First, observe that excess risk bound in Corollary 5 can be converted to a corresponding $T$-step regret bound by an application of Lemma 7 such that

$$
\mathfrak{R}_T \lesssim O\left(L_{\mathcal{K}}^{\frac{d}{\beta+2+2d}}\cdot T^{\frac{2\beta+4+4d}{3\beta+4+4d}}\right) .
\tag{67}
$$

For the class of Matérn kernels, the decay parameter $\beta = 1 + \frac{2\nu}{d}$ (Janz et al., 2020, Theorem 9). Using this wit the above regret bound, we get,

$$\mathfrak{R}_{\mathsf{Mat},T} \lesssim O\left(L_{\mathcal{K}}^{\frac{d^2}{2\nu+d(3+2d)}} \cdot T^{\frac{4\nu+d(6+4d)}{6\nu+d(7+4d)}}\right) .$$

This completes the proof. □

## D  ADAPTIVE SAMPLING VIA GP-UCB

In this section, we prove an upper bound on the expected risk of the Gaussian process upper confidence bound algorithm (GP-UCB) algorithm of Srinivas et al. (2010). In order to adapt their algorithm for our setup, consider the function

$$f_r(x) := \langle r, Mx \rangle_{\mathbb{H}_r} \quad \text{such that } D = \{x \mid \|x\|_{\mathbb{H}_\pi} \leq 1\}. \tag{68}$$

We have used $x$ to denote policies in this setup to be consistent with the notation in Srinivas et al. (2010). Observe that the domain defined above is not compact – a necessary condition for the algorithm to work. One work around this is to truncate the unit ball after a finite number of dimensions and bound this truncation error. The excess risk incurred by this truncation can be made arbitrary small. Going forward, we ignore this truncation. The regret for the UCB algorithm is shown to be upper bounded by $\tilde{O}(\gamma_T \sqrt{T})$ where $\gamma_T$ is the information gain with

$$\gamma_T := \max_{x_1,\ldots,x_T \in D} \frac{1}{2} \log \det(I + [\mathcal{K}(x_i,x_j)]_{i,j=1}^T) , \tag{69}$$

where we have assumed without loss of generality that the noise variance $\tau = 1$. For our setup, the kernel function $\mathcal{K}(x_i,x_j) = \langle Mx_i, Mx_j \rangle_{\mathbb{H}_r}$. We additionally require that the reward function $r$ belongs to the RKHS spanned by the set $\{Mx \mid x \in D\}$. Denote by $S = S_\pi^{\frac{1}{2}} M^\top S_r^{-1} S_\pi^{\frac{1}{2}}$ and suppose that its eigenvalues satisfy a power law decay with $\sigma_j(S) = \zeta_j = j^{-\beta}$. The following lemma upper bounds the information gain for this setup in terms of the power law parameter $\beta > 0$.

**Lemma 8** (Information Gain.)**.** *The information gain $\gamma_T$ for the above setup is bounded as*

$$\gamma_T = O(\log(T) \cdot T^{\frac{1}{\beta+1}}) . \tag{70}$$

*Proof.* The quantity of interest here is the information gain

$$\gamma_T := \max_{x_1,\ldots,x_T} \frac{1}{2} \log \det(I + XSX^\top) \quad \text{such that } \forall j \, \|x_j\|_2 \leq 1 , \tag{71}$$

where the matrix $X = [x_1^\top; \ldots; x_T^\top]$ and we have assumed that the noise variance is 1. From the setup described above, we have that the eigen values of $S$ decay as $\lambda_j \asymp j^{-\beta}$. It is easy to see that

$$F_{\mathsf{ig}}(\{x_t\}) := \frac{1}{2} \log \det(I + XSX^\top) \tag{72}$$

is a monotonic sub-modular function. Thus, the value of $\gamma_T$ can be upper bounded by $(1 - 1/e)^{-1}$ times the value of the greedy maximization algorithm. The greedy maximization algorithm is equivalent to picking

$$x_t = \arg\max_x F_{\mathsf{ig}}(X_{t-1} \cup \{x\}) .$$

It is easy to see that at each time $t$, the unit vector $x_t$ will be an eigen vector of the matrix $S$. Given this observation, we can finally upper bound the value of the info gain

$$\gamma_T \leq c \cdot \max_{m_1,\ldots,m_T} \sum_{j=1}^T \log(1 + m_j \lambda_j) \quad \text{such that } m_j \geq 0 \text{ and } \sum_j m_j = T.$$

Solving the above optimization problem, the optimal choice of the variables

$$m_j = \max\left\{\frac{1}{\lambda} - \frac{1}{\lambda_j}, 0\right\} \quad \text{and} \quad \sum_j m_j = T . \tag{73}$$

Setting $\lambda = T^{-\frac{\beta}{\beta+1}}$ ensures that there are $T^{\frac{1}{\beta+1}}$ active directions. Substituting the above values of $m_j$ in the expression for $\gamma_T$, we get

$$
\begin{aligned}
\gamma_T &\leq c \cdot \sum_{j=1}^{\infty} \log(1 + \max(\frac{\lambda_j}{\lambda} - 1, 0)) \\
&\leq c \cdot \log\left(\frac{\lambda_1}{\lambda}\right) \cdot \sum_{j=1}^{\infty} \mathbb{I}[\lambda_j > \lambda] \\
&\overset{(i)}{=} O(\log(T) \cdot T^{\frac{1}{\beta+1}}) \,,
\end{aligned}
$$

where (i) follows from setting $\lambda = T^{-\frac{\beta}{\beta+1}}$. This establishes the required claim. $\qquad\square$

We are now ready to state this our sample complexity bound for GP-UCB for this subclass of problems.

**Proposition 2** (Sample complexity for GP-UCB). *Suppose that the police space $\mathbb{H}_\pi$, reward space $\mathbb{H}_r$ and the map $M$ satisfy the power law decay assumption with exponent $\beta > 0$. The estimator $\hat{\pi}_{\mathsf{ucb}}$ output by the GP-UCB algorithm satisfies*

$$
\mathbb{E}[\Delta(\hat{\pi}_{\mathsf{ucb}}; r^*)] \leq \tilde{O}(n^{-\frac{\beta-1}{2(\beta+1)}}) \,. \tag{74}
$$

The proof of the sample complexity bound in Proposition 2 now follows the regret bound of $\tilde{O}(\gamma_T \sqrt{T})$ along with using the upper bound on the information gain from Lemma 8.

$$
\mathbb{E}[\Delta(\hat{\pi}_{\mathsf{plug}}; r^*)] = \tilde{O}(n^{\frac{1}{\beta+1}-\frac{1}{2}}) = \tilde{O}(n^{-\frac{\beta-1}{2(\beta+1)}}) \,. \tag{75}
$$

More recently, Cai & Scarlett (2021) extended the analysis of Valko et al. (2013) to show that the SupKernelUCB algorithm achieves a regret bound $\tilde{O}(\sqrt{\gamma_T T})$. Using this modified bound, one can improve the above analysis to obtain excess risk

$$
\mathbb{E}[\Delta(\hat{\pi}_{\mathsf{plug}}; r^*)] = \tilde{O}(n^{\frac{1}{2(\beta+1)}-\frac{1}{2}}) = \tilde{O}(n^{-\frac{\beta}{2(\beta+1)}}) \,, \tag{76}
$$

which is still worse than those obtained by the bounds by our proposed ridge regression estimator.

## E    FURTHER DETAILS ON EXPERIMENTAL EVALUATION

In the simulation study, we work with $d$ dimensional RKHSs $\mathbb{H}_r$ and $\mathbb{H}_\pi$. In order to simulate the nonparmeteric regime, we typically use value of $n$ which are less or at most a constant times the dimension $d$. We set the matrices $S_\pi = \mathrm{diag}(j^{-1.75})$, $S_r = \mathrm{diag}(j^{-1})$ and the map $M = I$. This is allowed since the policy space is smaller than the reward space. With this, the effective decay parameter $\beta = \beta_\pi - \beta_r = 0.75$. We sampled the true reward $r^*$ uniformly at random from the unit ball in $\mathbb{H}_r$. We further sampled the oracle noise $\epsilon \sim \mathcal{N}(0, 0.01)$. All plots were averaged over 10 runs.

