# OpenReview forum: "Reward Learning as Doubly Nonparametric Bandits:  Optimal Design and Scaling Laws"
_ICLR.cc/2022/Conference — ICLR 2022 Submitted_

### Official Review · Reviewer_qTko · 2021-10-30

**Correctness:** 4
**Technical Novelty And Significance:** 3
**Empirical Novelty And Significance:** Not applicable
**Recommendation:** 6
**Confidence:** 4

**Main Review:**

Overall I feel this is an interesting work and well-written. Section 4 has its value. I have some specific comments below.

1. I feel the setting is very close to the study of simple regret in the pure exploration problem in the bandit community. (2) is essentially the simple regret? I feel it's a bit weird to call (1) an oracle. It's just bandit feedback? Maybe the authors are from other communities but I hope you could relate or comment with bandits language in Section 2. Pure exploration with a simple regret guarantee is a well-studied area where you could refer Bandit Algorithm book.

2. Some important references on GP-bandits are missing. The minimax optimal regret for the Matern kernel of GP-bandits has been solved recently. The rate is T^{(\nu+d)/(2\nu+d)}. The upper bound appears in "On Information Gain and Regret Bounds in Gaussian Process Bandits, AISTATS 2021" and the lower bound appears in "On Lower Bounds for Standard and Robust Gaussian Process Bandit Optimization, ICML 2021". The regret bound in this paper is clearly sub-optimal when reducing to GP-bandits. Please correct me if I am wrong. If this is correct, I think it is very important to discuss how the current result can go beyond the GP-bandits setting and how important the nonparametric policy is. And how the current regret bound explains the hardness of more general policy class.

3. I feel it should be cautious to argue passive learning is better than adaptive learning in terms of simple regret. The bound you compare with is not sharp. Actually, in a recent work (Bandit Phase Retrieval, https://arxiv.org/abs/2106.01660), the authors have shown that adaptive learning is strictly better than passive learning (the rate is sharp there). Their model is a strictly sub-class of your model I think.

4. In Section 5, do you require the space of input points to be the full unit ball? Because when you convert your algorithm into an online regret minimization algorithm, the explore-then-commit should not be a good one unless your action set has some good curvature to use, like the full unit ball.

5. How it relates to optimal design? Indeed, in the main section, the word "optimal design" does not even appear. If it appears in the title, I feel you should explain what do you mean by optimal design explicitly.

**Summary Of The Paper:**

This paper essentially studies pure exploration in nonparametric bandits setting and provides a simple regret guarantee. The authors generalized the setting to nonparametric policy class. Some new ideas based on minimizing a risk upper bound are proposed.

**Summary Of The Review:**

This paper presents some interesting ideas and generalizes the setting to nonparametric policy class. I hope the authors could clarify their contribution w.r.t the SOTA rate.

---

> ### Author Response · Authors · 2021-11-18
> **Author response**
>
> We thank the reviewer for their detailed comments as well as for pointing out recent related work which we had missed out on. We have updated our draft to incorporate all the missing references. Below, we answer the concerns raised.
>
> > *“simple regret in the pure exploration problem ”*
>
> Thanks a lot for pointing this out, we have added a discussion on the connection between the two naming conventions.
>
> > *“Some important references on GP-bandits are missing. Important to discuss how the current result can go beyond the GP-bandits setting and how important the nonparametric policy is”*
>
> We have added in the missing references and have included a comparison of our result for the nonparametric policies with these results in Appendix D. Overall, the simple regret bounds obtained by our proposed method are better than those obtained by these improved analyses as well for our doubly non-parametric setup. While the bounds of the stated paper are minimax regret bounds for the kernel MAB problem, especially the Matern kernel, our simple explore-then-commit approach yields non-vacuous regret bounds in scenarios where most of the existing and practically used algorithms (GP-UCB, GP-TS) fail.
>
> > *“be cautious to argue passive learning is better than adaptive learning in terms of simple regret.”*
>
> We completely agree that more generally, we expect adaptive learning to give better bounds than those obtained by passive algorithms. This is indeed a direction we are exploring ourselves. Our claims in section 4.3 pertain to a particular adaptive algorithm, that based on UCB style approaches and our upper bounds indicate that our passive learning algorithm might be better than these. We have updated the wording of our claims in this section to add more clarity.
>
> > Unit ball assumption in Section 5
>
> For the case of Section 5, we do consider the input domain space $\mathcal{X}$ to be the unit ball in $d$ dimensions. Our analysis goes through even if the space is not the entire unit ball but a subspace of it. As you pointed out, however, this might not give optimal rates when we consider the explore then commit style algorithm on top of our passive strategy.
>
> > *“How it relates to optimal design?”*
>
> Our framework requires the learner to select a set of query policies and algorithmically, we show that this is equivalent to studying the optimal design problem for a certain ridge regression problem. More precisely, the optimal set of query policies for our learner are obtained by optimizing the upper bound in equation (5) in our paper. This is precisely the problem of optimal experiment design for the ridge estimator.

---

### Official Review · Reviewer_oTMH · 2021-10-30

**Correctness:** 4
**Technical Novelty And Significance:** 4
**Empirical Novelty And Significance:** 4
**Recommendation:** 6
**Confidence:** 2

**Main Review:**

This is a solid contribution in an impressively general setting, which I believe be interesting to many working in this area and inspire future work. The challenge is well motivated in the introductory sections and the theoretical results show the proposed approach has strong performance. I have not been able to fully check all of the mathematical work in the appendices, but what I have inspected seems to be accurate and non-trivial. While I am assigning a positive score, I think the exposition around the improvement over UCB-style algorithms could be improved.

Presently, there is a comment saying that you believe the improvement in terms of the theory is due to the proposed approach being a better approach rather than some gap in the theory. This doesn’t feel as strong as it could be – could you supplement this with some more details as to what features make the difference? It is, as you identify, quite a surprising result that popular adaptive algorithms are theoretically outperformed by passive approaches. Could some of the elements of this passive approach be employed to produce a yet stronger adaptive approach in future work do you think?

Minor Comments
•	Typo near the bottom of page 2: “GP-UCB and GP-TS are only yield sub-linear”
•	Typo in second para of Section 3: “Such general plug-in procedure have”


**Summary Of The Paper:**

This paper is motivated by learning optimal actions in tasks where both the reward function, and policy (actions) are nonparametric. Previous literature has typically only considered one of these two components as being nonparametric. The main focus is on reliably identifying a policy with low instantaneous regret/risk via as few queries as possible, where all queries are specified in advance (i.e. the passive query setting). The proposed approach selects query locations based on an eigenvalue decomposition of the policy space, sampling repeatedly along a set of top eigenvectors to ensure reliable estimation of the reward via a plug-in ridge-regression-based regression. This estimated reward function is then minimised over the policy class to return a suggested policy/action. The accompanying theory shows the decay of the risk of this suggested action, and shows that in the setting of the Gaussian Process bandit, this can enjoy a better rate than existing adaptive approaches such as GP-UCB.

**Summary Of The Review:**

I am positive about this submission, I think it is interesting, innovative and potentially very impactful. The results are impressive, though I think there is an opportunity to give a bit more insight as to the conceptual reasons for the more surprising aspects of the results.

---

> ### Author Response · Authors · 2021-11-18
> **Author response**
>
> We thank the reviewer for their detailed feedback and kind comments. We address the concerns raised in the review below.
>
> > *“exposition around the improvement over UCB-style algorithms could be improved”*
>
> We have updated Section 4.3 of our manuscript with more details and intuition into why our approach performs better than typical UCB style algorithms.
>
> > Comparison with UCB algorithms
>
> UCB style algorithms require the construction of confidence intervals around input points, which crucially dictate the regret bounds of such algorithms. In the frequentist setup, the best known such bounds are known to yield suboptimal regret rates and it is an open question as to whether these can be improved. Intuitively, these large confidence intervals make the algorithm explore significantly more than what is required and are responsible for their sub-optimality. In contrast, our algorithm goes via a ridge regression route and for the passive setup, is able to optimize the excess risk by optimally trading off the bias and variance.
>
> > *“elements of this passive approach be employed to produce a yet stronger adaptive approach in future ”*
>
> This is a really interesting suggestion and one which we have been working on as a research problem. As also pointed out in the response to Reviewer kffS, we believe that our approach might be optimal in a worst-case scenario but not in an instance specific manner. We believe that an adaptive approach which builds on top of our passive approach can perhaps yield better instance dependent results which we are looking into.

---

> > ### Comment · Reviewer_oTMH · 2021-11-20
> > **Reply to Author Response**
> >
> > Thanks for providing these clarifications and updates. I will keep my positive score.

---

> > > ### Comment · Reviewer_oTMH · 2021-12-06
> > > **Follow up to reviewer discussion**
> > >
> > > Having reflected on the comments of other reviewers with greater confidence scores and their responses to the author discussion, I will move my score to a 6, as I this more moderate position now better represents my opinion. I still think the paper is nicely put together, but the issues that the other reviewers raise do seem to be valid concerns also.

---

### Official Review · Reviewer_k2hj · 2021-11-03

**Correctness:** 4
**Technical Novelty And Significance:** 2
**Empirical Novelty And Significance:** 1
**Recommendation:** 5
**Confidence:** 3

**Main Review:**

The proposed techniques are shown to yield non-asymptotic excess risk bounds for a simple plug-in estimator based on ridge regression. The general results can be applied to the specific settings of GP-Bandits and are shown to yield competitive regret guarantees for the Matern kernel.

The paper is readably well written and the proofs are sound which is however mostly borrowed from the existing analysis of GP-Bandits (e.g. Srinivas, 2010).

One concern with the problem formulation (doubly-nonparametric bandits) is it seems to be closely tied to the framework of GP-Bandits which is well studied in the literature, it is unclear the scopes and motivations of the proposed frameworks beyond that are already covered by GP-Bandits. Can you point out a real-world problem that can not be resolved under the GP-Bandits framework but with a doubly non-parametric bandits framework? Due to the same reason, the proposed ridge regression-based algorithms are also lifted from standard methods used in GP bandits as well as the analysis techniques. So it is hard to appreciate the specific novelties of this paper, or the unique contribution that is missing the earlier works. This does not reflect from the contribution section of the paper as well.

Any comments on the computational complexity of the proposed algorithm given any arbitrary non-parametric reward and policy class?
Finally, the experimental evaluation section of the paper is extremely weak, there have been no comparisons made with state-of-the-art methods, even the algorithms of Kernelized-MAB as listed in Table 1 which is surprising.

The paper has some minor typos, e.g. “doubly-nonparameteric” in Pg 2, please proofread the draft thoroughly. A separate problem formulation and technical contributions section would also be helpful for the readers.


**Summary Of The Paper:**

The paper addresses the problem of reward learning by learning a reward model from human feedback using the optimal design of the queries. Authors address this by essentially framing the problem in the flavor of GP-Bandits that models rewards and policies as non-parametric functions belonging to subsets of Reproducing Kernel Hilbert Spaces (RKHSs) where the learner receives (noisy) oracle access to a true reward and is expected to output a near-optimal (reward maximizing) policy. More precisely they analyze the framework of doubly-nonparametric bandits for theoretically studying the reward learning problem.


**Summary Of The Review:**

The theoretical findings of the paper are sound but it is hard to appreciate the novelties of this work over the existing techniques and analysis of GP-Bandits. I will be happy to increase the score if authors can precisely point the new challenges overcome by this work and what the one novel idea unique to this work.

---

> ### Author Response · Authors · 2021-11-18
> **Author response**
>
> We thank the reviewer for their thoughtful feedback and address the concerns below.
>
> > *“The paper is readably well written and the proofs are sound which is however mostly borrowed from the existing analysis of GP-Bandits”*
>
> While our analysis for the fixed design setup borrows tools from the analysis of ridge regression, our overall set of techniques are very different from the ones used in the GP bandit literature. Because of the sequential nature of the task, most works focus on bounding the information gain across a set of T timesteps (see Section 4.3 and Appendix D of our paper). In contrast, our analysis proceeds via a bias-variance analysis of a ridge regression procedure and understanding the properties of the optimal covariance matrix.
>
> > *“A separate problem formulation and technical contributions section would also be helpful for the readers”*
>
> Section 2 in our paper formalizes the problem formulation. We have added more technical details to our contributions paragraph at the end of Section 1.
>
> > *“(doubly-nonparametric bandits) is it seems to be closely tied to the framework of GP-Bandits”*
>
> Our doubly-nonparametric bandit framework actually differs from the commonly studied GP-Bandit framework in allowing the set of query policies to be a non-compact subset of an RKHS. This allows one to reason about deep neural network based rewards and policies which can be viewed as non-parametric functions. For instance, our framework allows both rewards and policies to be appropriate subsets of classes of smooth functions, say induced by different Matern kernels, and Theorem 1 then provides sufficient conditions for learnability for this setup.
>
> > Computational complexity
>
> The exact computational complexity of the proposed ridge regression estimator depends on how the particular functions in the RKHS are represented. Assuming blackbox access to the map M, and $O(1)$ access to functions in the RKHS, one can run the ridge regression algorithm in time $O(n^3)$ required to invert the appropriate kernel matrix.

---

### Official Review · Reviewer_kffS · 2021-11-05

**Correctness:** 4
**Technical Novelty And Significance:** 2
**Empirical Novelty And Significance:** 2
**Recommendation:** 5
**Confidence:** 4

**Main Review:**

Technical correctness:

I did not check the proofs in detail. As far as I see the results are consistent with the literature and I do not have any particular concerns.

Clarity:

The writing is technically reasonable (see the minors for some exceptions). The paper does not help the reader very much developing intuitions. You could write, for example, what the algorithm is doing in the finite-dimensional linear setting. At the same time, I would encourage the authors to spell out where is the novelty in their analysis. What technical barrier has been overcome? What parts of the proof would be most interesting and most useful in future analysis?

Novelty:

I am not a super expert on kernel methods for bandits. I am a little surprised that the basic results are new. I suppose experimental design is already a little bit niche. Anyway, someone more expert than me should probably comment on the novelty.

Other comments:

(1)

It is interesting that an improvement in the rate of the cumulative regret is possible relative to UCB. The author could explain a little more why this is possible. My own guess is that the non-sequential way the data is collected leads to an improved dependence on the effective dimension, which in the kernel setting is coupled to the horizon. This effect appears in the finite-dimensional setting where the phased algorithms achieve a bound of sqrt(d T log(k)) with k the number of actions and d the dimension. UCB one obtains only d sqrt(n).

Note that explore-then-commit in this setting yields a suboptimal T^{2/3} rate, but the dimension dependence is improved by using the non sequential estimators.

An obvious question this raises is whether or not a phased elimination algorithm using your risk bounds can improve even further. The algorithm I am proposing would operate in phases m=1,2,... with the number of interactions in phase m being 2^m. Within each phase the algorithm would use the same procedure as your simple regret algorithm to collect data and then eliminate policies based on confidence intervals.

Does this idea lead to an even better rate? If not, can you explain why? Note you will need high probability confidence intervals for this. But I guess you are (could) derive them already for your analysis.

Note that some previous work also handles the contextual bandit version of this problem, which is not possible using explore-then-commit.

(2)

Another obvious question is whether or not you can get high probability bounds in Theorem 1. I guess the answer is yes.

(3)

There is a related literature on best arm identification in linear bandits. Here the algorithms are generally adaptive and the
bounds depend on the relationship between the policies (actions). For example, [1] below, but there are many more recent papers (see citations to this paper). While these works often study finite (dimensional and action) settings, they do have a problem-dependent nature. It would be interesting to investigate this possibility here.

[1] Soare et al. Best-Arm Identification in Linear Bandits (2014).

(4)

My last comment is on Assumption 1. How benign is this? And how necessary is it? In the finite-policy linear setting the assumption in general simply does not hold. What is interesting there is that by solving the log-det problem you can nevertheless find a design that yields the same minimax rate independent of the geometry of the policy set. Surely we should wonder if the same is true here. My intuition says yes, since by introducing the effective dimension everything becomes finite. In summary: can we obtain the same results without this assumption by first introducing the effective dimension and then solving a standard D-optimal design problem? If not, why not? If so, why not do it?





Minors:

* P2. The regret has not been defined in Table 1.

* P3. "pi^* \in argmax ..." but why should this exist? Some compactness assumption?

* P3. The coefficients mu_pi and mu_r were not explained\introduced.

* P4. "policy Learning via reward learning" -> "policy learning via reward learning"?

* P4. There is a missing expectation in the RHS of (7).

* P5. I am not sure if the singular values sigma_j are defined (or I missed where)

* P5. It was not clear to me whether or not the assumption in (9) will be used for the remainder.

* P5. J is introduced abruptly and before it has really been defined.

* P6. "where the quantity zeta_j = ... for some universal constant c > 0" -> "where the quantity zeta_j = ... and c > 0 is a universal constant."

* P6. On what does c_pi depend? I think it is a universal constant but the order of quantifiers in the assumption is a bit confusing.

* P6. "for instance via convexification" -> can you explain this more?

* P7. "does not flip the larger eignevectors of ..." more explanation would be helpful here as well.

* P8. "we let N_cov(eps) denote an eps-net of". I guess really N_cov(eps) is the size of an epsilon net. The net itself seems to be C defined in the next sentence? Or this terminology is unfamiliar to me.

* What you call policies are perhaps more commonly called actions.

* What lower bounds do we have in these settings?

* The observation that rates improve when the policy set is the unit ball was observed (in the finite-dimensional setting) by Rustemevichientong et al. Linearly Parameterized Bandits (2010).


**Summary Of The Paper:**

The paper studies a pure exploration bandit problem in a setting where the reward and policies are assumed to
be in RKHS spaces. The main result is a passive strategy based on experimental design and bounds on
the simple regret. The results are extended to the cumulative regret setting via a standard explore-then-commit argument.
In the latter setting the results are (depending on the kernel) about competitive with the state-of-the-art (more on this later).

**Summary Of The Review:**

The paper executes about the first thing you would try. This can be a strength and a weakness. I wish the authors provided more insight in their writing. Not only explaining what holds, but also why and putting in the context of existing work. The paper could be made stronger by investigating any or all of the directions suggested in (1)-(4) above.

---

> ### Author Response · Authors · 2021-11-18
> **Author response**
>
> We thank the reviewer for their thoughtful comments. Below, we answer the concerns raised in the review.
> > Clarity of writing
>
> We have corrected all the typographical errors pointed out in the minor points in the updated draft and have also added intuition regarding the finite dimensional case above Proposition 1.
>
> > Novelty in analysis
>
> We agree that the analysis for the fixed design setup is standard and based on the bias-variance analysis for ridge regression. However, our work goes beyond this standard analysis and our contributions are:
> * From a conceptual standpoint, our work contributes by formalizing the reward learning problem as a doubly-nonparametric bandit problem. This formalism allows one to understand the practical issues including selection of good query policies as well as understanding how the respective size of policy and reward class affect learning.
> * From a technical perspective, given the fixed design analysis, we still need to choose good query points. A key insight for identifying good queries is that the excess risk depends only on the properties of the operator $S_\pi^{\frac{-1}{2}}M^\top S_r S_\pi^{\frac{-1}{2}}$ as well as the empirical covariance matrix $\Sigma_n$.
> * Reduction of kernel multi-armed bandits to our framework. Our reduction carefully constructs two different RKHS to embed the policy and reward class based on a covering of the input ball. This is a non-standard construction and our regret bounds in Theorem 3 show that it gives non-trivial results for the kernel MAB problem.
>
> > Surprised that basic results are new
>
> We agree with the reviewer that it was indeed surprising that these basic results were previously unknown. One reason is that since the formal study of the problem by Srinivas et al. the focus of the community has largely been around the adaptive case and the corresponding passive (or simple regret) case has received much less attention in the past decade.
>
> > Comparison with UCB algorithms
>
> UCB style algorithms require the construction of confidence intervals around input points, which crucially dictate the regret bounds of such algorithms. In the frequentist setup, the best known such bounds [1] are known to yield suboptimal regret rates and it is an open question as to whether these can be improved. Intuitively, these large confidence intervals make the algorithm explore significantly more than what is required and are responsible for their sub-optimality. In contrast, our algorithm goes via a ridge regression route and for the passive setup, is able to optimize the excess risk by optimally trading off the bias and variance.
>
> > Phased Elimination strategy
>
> This is a very good idea and we did think about this to convert our passive to an active strategy. There were a couple of challenges in obtaining better rates from this. In order to use a phased elimination strategy, one would need to construct valid confidence intervals on the policy set, which as mentioned above, worsens the regret bound by a considerable margin. Second, and more importantly, since our setup allows for infinite dimensional policies, it wasn’t clear a apriori whether such a strategy would be useful in the worst-case scenario since there will always be directions one cannot explore in finite samples. This does raise open a very interesting question on understanding the specific structure on true reward $r^*$ under which this strategy can perhaps give a better rate.
>
> > High probability bound for Theorem 1
>
> Thanks for suggesting this. We have updated the draft to also include the high probability bound in the appendix.
>
> > Instance dependent rates
>
> Thanks a lot for pointing out this paper. Studying instance dependent rates for our proposed formulation is indeed an interesting yet technically challenging problem. We plan to explore this as future research since it requires quite a different set of analysis techniques than those presented in the paper.
>
> > Regarding Assumption 1
>
> Our results hold more generally under weaker conditions for the admissible policy set -- we only require that there exists policies in the set such that one can obtain a covariance matrix which dominates (in psd ordering) the optimal one by a constant factor. Assumption 1 is a sufficient condition to ensure this.
>
> > *"In the finite-policy linear setting the assumption in general simply does not hold."*
>
> The finite policy case is very different from the infinite one with regards to this assumption. With a finite set (in fact in finite dimensions), one can actually explore all the policies (or directions) with a large enough sample size and get around this issue. On the other hand, in the infinite dimensional setup, there will always exist certain directions which the algorithm cannot explore.
>
>
> **References**
>
> [1] Vakili, S., Khezeli, K. and Picheny, V., 2021, March. On information gain and regret bounds in Gaussian process bandits. In International Conference on Artificial Intelligence and Statistics.

---

### Author Response · Authors · 2021-11-22
**Post-rebuttal response**

Dear Reviewers,

We would really appreciate if you could you please take a look at our responses to your concerns and let us know if we have appropriately answered all your questions. We are happy to engage and clarify any further questions that you may have on those concerns. Since today is the response deadline, we will not be able to make any changes to our submitted manuscript after this.

Thanks.

---

### Decision · Program_Chairs · 2022-01-20

**Decision:**

Reject

**Comment:**

While the reviewers found several interesting points about the paper, they raised several issues, which prevents me from recommending acceptance of the paper. In particular, the paper is not positioned properly in the literature, hence the novelty and the contributions are not properly clarified. The approach of the paper is reasonably simple (which would be a good thing by itself), but there seem to be natural avenues along which more complete results could be obtained, as mentioned in the reviews. Finally, the experiments should be improved (e.g., comparing with other algorithms from the literature). In summary, this is a promising work, but it requires some improvements before it can be published.